# DAS-GNN: Degree-Aware Spiking Graph Neural Networks for Graph Classification

## Abstract

The recent integration of spiking neurons into graph neural networks has been gaining much attraction due to its superior energy efficiency. Especially because the sparse connection among graph nodes fits the nature of the spiking neural networks, spiking graph neural networks are considered strong alternatives to vanilla graph neural networks. However, there is still a large performance gap for graph tasks between the spiking neural networks and artificial neural networks. The gaps are especially large when they are adapted to graph classification tasks, where none of the nodes in the test set graphs are connected to the training set graphs. We diagnose the problem as the existence of neurons under starvation, caused by the sparse connections among the nodes and the neurons. To alleviate the problem, we propose DAS-GNN. Based on a set of observations on spiking neurons on graph classification tasks, we devise several techniques to utilize more neurons to deliver meaningful information to the connected neurons. Experiments on diverse datasets show significant improvements compared to the baselines, demonstrating the effectiveness of the DAS-GNN.

## 1 Introduction

Graph neural networks (GNNs) are popular neural networks that learn representations from graphs, which comprise multiple nodes and edges between them. Because of their flexibility to model any kind of connection that exists in nature, they have various applications ranging from drug discovery (Borgwardt et al., 2005; Wale et al., 2008; Debnath et al., 1991), social influence prediction (Qiu et al., 2018; Arazzi et al., 2023), traffic forecasting (Bai et al., 2020; Cao et al., 2020), and recommendation systems (Pal et al., 2020; Fan et al., 2019; Zhang et al., 2023). One known challenge of GNNs is their sparse memory and computational pattern. Because many messages are passed between randomly connected nodes, there is a significant inefficiency in processing them with conventional systems (Yan et al., 2020; Yoo et al., 2022; 2023; Geng et al., 2020).

To address the inefficiency, spiking neural networks (SNNs) are considered strong alternatives. Inspired by the biological behavior of brains, SNNs process information by communicating binary spikes between the neurons. Because SNNs utilize intermittently occurring spikes, they have superior energy efficiency, especially in the domain of GNNs (Aimone et al., 2021).

Although the spiking graph neural network (SGNN) has been recently studied by many researchers (Li et al., 2023; Zhu et al., 2022; Wang & Jiang, 2022), we observe huge performance drops when adapted to graph classification, compared to that of the conventional GNNs implemented with artificial neural networks (ANNs). Upon closer analysis of the performance degradation, we identify spike frequency deviation of the neurons within the model. In our investigation, many neurons experience *starvation*, which does not emit any spike during the inference. This leads to severe information loss, due to being unable to deliver signals to the subsequent neurons.

Such a problem was less exposed in previous studies on spiking GNNs. This is because the test set nodes are available during the training time (transductive learning (Kipf & Welling, 2016)) or they are part of the training graph (inductive learning (Hamilton et al., 2017)). In such settings, the model could be trained on the nodes to mitigate the performance drop. However, in graph classification tasks, the graphs are independent of each other, and the test set comprises multiple unseen graphs, aggravating the problem.

Fortunately, our further analysis reveals that such phenomena are related to the degree of the vertices in input graphs. We discover that a strong pattern exists among the neurons in the GNN, where 1) neurons in a node have similar behaviors, 2) each feature dimension presents different behaviors, and 3) neurons in high-degree nodes tend to emit more spikes.

Motivated by the observations, we propose to group the neurons according to the degree of the node (*degree-aware group-adaptive neurons*). The neurons in each group adapt the threshold voltage together to steer the firing rate toward ideal rates. To further improve the neuron groups being sensitive to the base threshold value at the first inference step, we further propose to learn the base threshold voltage values (*learnable inference base threshold*).

We evaluate DAS-GNN over multiple GNN models and datasets. Experiments reveal that DAS-GNN achieves superior performance over the SNN baselines, setting a new state-of-the-art method for SNN-based graph classification. Compared to the ANN counterparts, we demonstrate that DAS-GNN has advantages in energy efficiency, similar to what has been reported in many SNN-based approaches. Notably, contrary to the usual studies where ANNs serve as performance upper bounds for SNNs, we observe several cases where DAS-GNN significantly outperforms the ANN counterparts. This suggests that GNN-based graph classification might be a field where SNNs could excel.

Our contributions are summarized as the following:

- We identify the starvation problem of spiking neurons in GNNs for graph classification tasks that harms the performance. We further observe that the spike frequency patterns have a strong correlation with the vertex degree.
- Based on the observations, we propose degree-aware group-adaptive neurons, which dynamically adjusts the threshold voltage together with the other neurons in the group to address the spike frequency deviations.
- We propose techniques to reduce the sensitivity of the neuron groups to the inference base threshold at the first step, by learning the base threshold for each group.
- We evaluate DAS-GNN on several public graph classification datasets, and the experimental results show that DAS-GNN achieves superior performance over existing techniques.

## 2 BACKGROUND

### 2.1 SPIKING NEURAL NETWORKS AND SPIKE TRAINING

Spiking neural networks (SNNs) are third-generation neural network designs that mimic human biological neural systems (Maass, 1997). They use spike-based communication and adopt event-driven characteristics that promote better energy efficiency than current ANNs. Similar to human neural systems, SNNs consist of spiking neurons that can model spatio-temporal dynamics of the actual biological neurons. The early forms of such neuron models are Hodgkin-Huxley neurons (Hodgkin & Huxley, 1952), which accurately model the biophysical characteristics of the membrane through differential equations. However, its mathematical complexity prohibits its practical use and scalability. Instead, the Leaky Integrated-and-Fire (LIF) model finds a middle ground between mathematical simplicity and biological plausibility, and is popularly adopted as the baseline architecture (Hodgkin & Huxley, 1952). In the LIF neuron, the weighted sum of input spikes is accumulated over time within the neuron as membrane potential, and the output spike is generated only when the membrane potential exceeds a threshold value. This process is represented as a differential function:

$$\tau \frac{dU(t)}{dt} = -U(t) + I(t), \tag{1}$$

where $U(t)$ denotes the membrane potential value at time $t$, $\tau$ is a time constant of the membrane, and $I(t)$ is the input at time $t$ from connected synapses. To make this time-varying function computationally feasible, we discretize and rewrite it in an iterative form for a sequential simulation:

$$U(t) = U(t-1) + \beta(WX(t) - (U(t-1) - R)), \tag{2}$$
$$U(t) = U(t)(1 - S(t)) + RS(t), \tag{3}$$

$$S(t) = \begin{cases} 1, & \text{if } U(t) \geq V_{th} \\ 0, & \text{otherwise,} \end{cases} \tag{4}$$

where $\beta$ is a decay rate constant, $R$ is the reset value and $V_{th}$ is the threshold for the membrane potential. Note that $I(t)$ is simplified as weighted input $(WX(t))$, which can be obtained through any operations with learnable weights including convolution, self-attention, or a simple MLP. We will denote this process of forwarding through LIF neurons as $SN(\cdot)$ in this paper.

**Direct SNN Training.** Since SNNs were first implemented through ANN-SNN conversion (Cao et al., 2015), various studies have aimed to address the accuracy degradation that occurs during the conversion from ANNS to SNNs (Han et al., 2020; Rueckauer et al., 2017; Hunsberger & Eliasmith, 2015; Sengupta et al., 2019). However, due to the non-differentiable step function in Equation (4), direct SNN training with backpropagation is prohibited.

To enable direct SNN training, many approaches have been proposed (Shrestha & Orchard, 2018; Bohte et al., 2002; Esser et al., 2015; 2016; Che et al., 2022; Wu et al., 2018; Deng et al., 2022) to bypass the step function. Recent research has demonstrated that directly training SNNs can yield competitive results by addressing the challenges posed by non-differentiability. To this end, our work focuses on directly training Graph SNNs. While exploring ANN-SNN conversion methods would be an interesting scenario, they target a different dimension from ours.

## 2.2 GRAPH NEURAL NETWORKS

Graph neural networks (GNNs) take graph-represented data as input, which consist of nodes and their connected edges $\mathcal{G} = (V, E)$, with node features $X \in \mathbb{R}^{|V| \times F}$ and optionally edge features $\mathbf{E} \in \mathbb{R}^{|E| \times D}$. The common GNN architectures follow a message passing paradigm (Gilmer et al., 2017), which learns node or edge representations by aggregating information from its neighboring nodes and updating the node features iteratively. Thus, a single forward of GNN layer consists of message passing and combination: $h_i^{(l+1)} = \phi(h_i^{(l)}, \bigoplus_{j \in \mathcal{N}(i)} \psi(h_i^{(l)}, h_j^{(l)}, e_{ij}))$, where $l$ and $i$ are indices for layer and node, respectively, and $\psi(\cdot)$ denotes the message passing function. After the features in the message are aggregated, the combination phase uses $\phi(\cdot)$ for feature updates. For graph convolutional network (Kipf & Welling, 2016), the overall process can be simplified as:

$$X^{(l+1)} = ReLU(AX^{(l)}W^{(l)}), \tag{5}$$

where the feature matrix is a concatenation of node features $X^{(l)} = [h_0^{(l)}||h_1^{(l)}||...||h_{(|V|-1)}^{(l)}]^T$, which is updated through iterations of message passing $(AX)$ and combination $(XW)$. After iterative updates of $X$ through the layers, the learned node or edge embeddings are passed through an additional classification layer for node-level or edge-level predictions.

**Graph Classification** follows the same node-wise message-passing framework to obtain node embeddings but appends a readout layer to turn them into a single graph embedding:

$$h_G = R(h_i^{(L)}|V_i \in \mathcal{G}), \tag{6}$$

where $R$ denotes the readout function. The readout function reduces the node features to a single embedding regardless of the number of nodes. This is due to the inductive nature of graph classification tasks where the number of nodes is not known in advance. While GNN layers focus on communicating features only from a local neighbor of a vertex, the readout layer considers the entire graph to generate global features. The obtained graph embedding is passed through a classification layer for predictions. Graph classification tasks usually hold more difficulty than node-level classification due to its inductive nature, where inference is done on unseen graphs and thus cannot utilize any graph-specific statistics from the train set. In this paper, we mainly focus on graph classification.

## 2.3 SPIKING GRAPH NEURAL NETWORKS

In this paper, we adopt conventional SNN designs where LIF neurons are connected through learnable weights, and apply them to the GNN framework (Zhu et al., 2022). As mentioned in Section 2.2, each GNN layer outputs an updated feature matrix $X^{(l+1)} \in \mathbb{R}^{|V| \times F}$. This is converted to spike representation through the spiking neurons $SN()$:

$$X^{(l+1)} = SN(AX^{(l)}W^{(l)}). \tag{7}$$

As the input $X^{(l)}$ and the output $X^{(l+1)}$ are in spike format, spike representation is maintained throughout the model.

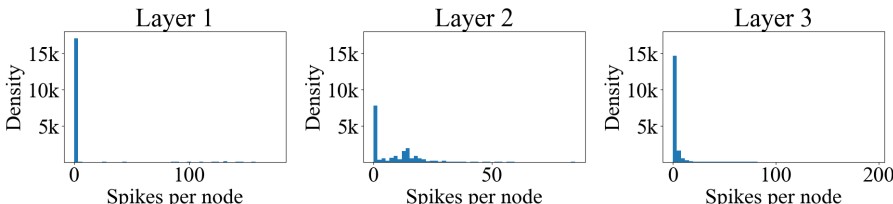

(a) Histogram plotting distribution of total spikes counted over time for each node. X-axis denotes spike counts from each node, while y-axis denotes density of each bin.

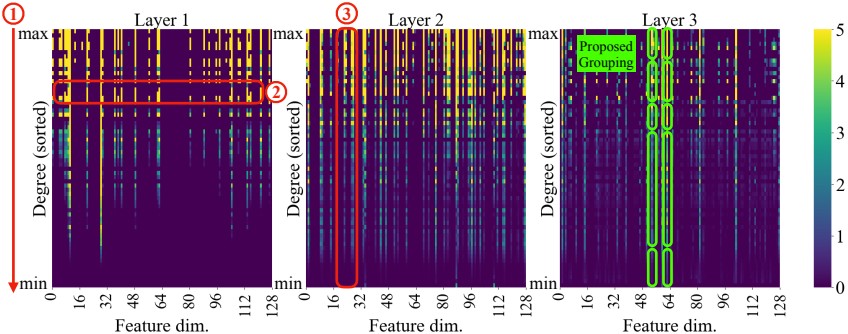

(b) Spike frequency visualization using each layer output. X-axis denotes feature dimension, while y-axis denotes nodes grouped and sorted by degree in descending order, top to bottom. Brighter spots denote higher spike frequency.

Figure 1: Analysis on spike frequency variation of GCN using IMDB-BINARY (Yanardag & Vishwanathan, 2015b) dataset.

## 3    ANALYSIS ON SPIKE FREQUENCY VARIATION OF GNNS

To analyze the cause of the accuracy drop, we plot the behavior of the neurons during inference in Figure 1a, on an IMDB-BINARY dataset over five timesteps ($T = 5$). We create a histogram of spike counts created from each node, which is associated with 128 neurons. As depicted in the plot, it is clear that most of the neurons are under starvation. This is caused by the inputs of those neurons being insufficient to reach the threshold, and this leads to severe information loss between the layers. While unveiling the exact dynamics would require more research, we hypothesize that this is caused by the varying degrees of vertices in real-world graphs.

To validate the hypothesis and further investigate the phenomena, we display the spike frequency heatmap of the neurons sorted by the degree of the nodes in Figure 1b. From the heatmap, we make three observations:

① **(Brighter on the top and darker at the bottom)** *High-degree nodes tend to exhibit higher spike frequencies.*

② **(The horizontal strips)** *The spike frequencies are associated with the corresponding nodes.*

③ **(The vertical strips)** *The feature neurons within a node behave differently according to their positions.*

We believe such patterns come from the connectivity of the nodes and the distinct role of the neurons assigned to each node. The connectivity affects the number of receiving spikes of neurons associated with each node. It is known that most of the real-world graphs exhibit an extremely skewed distribution of degrees (i.e., power-law distribution (Leskovec et al., 2007)). Due to such a characteristic, there are a few nodes with a very high degree, while a majority of nodes have a low degree. Because a GNN layer communicates signals between the neighbors through messages, a high-degree node will likely receive a lot of spikes, while a low-degree node will receive only a few.

These three observations shed light on how to close the performance gap between spiking GNNs and ANN-based GNNs. In the next section, we describe how the observations are used to build better spiking GNNs for graph classification.

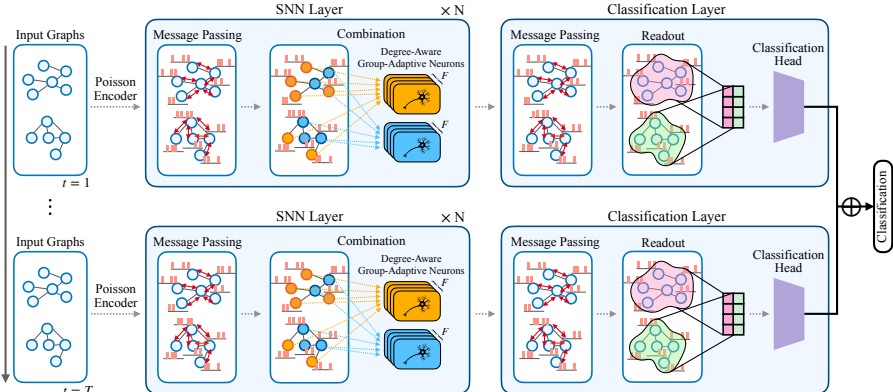

Figure 2: Overall graph classification architecture with proposed methods.

# 4 PROPOSED METHOD

## 4.1 OVERALL GRAPH CLASSIFICATION ARCHITECTURE

Many recent studies have tried to adapt SNN architectures into GNN tasks, however, they simply try to contact with only node classification tasks (Zhu et al., 2022; Wang & Jiang, 2022). In this work, we propose a spiking neural network specifically designed for graph classification tasks and show that it can be trained using spikes. We demonstrate the overall architecture of DAS-GNN in Figure 2. For each timestep, the input graphs are first translated into spike representations through the Poisson encoder, then the message passing is done in spike format. After the combination operation in the SNN layer, the node features are transformed into a spike format by passing the neurons. In the classification layer, we perform an extra message passing on the spike features before passing the readout layer. The readout operation reduces the multiple rows of node embedding by computing feature-wise averages, thus representing them into a single row to represent the entire graph. A batch of graph embeddings is passed through a classification head that outputs logits for each timestep. To make the final prediction, we simply take the sum of logits from all inference timesteps and use softmax to obtain the class probabilities.

## 4.2 DEGREE-AWARE GROUP-ADAPTIVE NEURONS

As discussed in Section 3, GNNs suffer from a huge gap in spike frequencies between neurons. As observed, there exists some patterns (Figure 1) that we can utilize to address the issue. One naive way of addressing the issue is to use learnable (Wang et al., 2022), or adaptive (Bellec et al., 2018) threshold for each neuron. By adjusting the threshold, one can expect the neurons to naturally change, such that neurons under starvation will have lower thresholds to fire more often, and a few neurons with high firing rates will have higher thresholds to shift toward an ideal distribution.

Unfortunately, such an idea cannot be directly applied unless all the testset nodes are available at training time (i.e., transductive task). However, such a setting would be considered a data leak for graph classification and would also lose the advantage SNNs have on lightweight inference. Moreover, the number of nodes in a real-world dataset often ranges from at least thousands to several billions. Considering that GNNs often involve only a sub-million number of learnable parameters, storing such a large number of thresholds is considered too much overhead.

To address the aforementioned issues, we propose *degree-aware group adaptive neurons* (DAG), which groups the neurons by their degrees and lets each group adapt its threshold. The neurons are split according to the degree of the associated vertex and further split along the feature dimensions. In Figure 1, we illustrate each group as vertical bars as depicted by the green boxes. Thus, we reformulate Equation (4) as below. In the equations, we focus on a single feature position, and use $N_g$ to denote the set of neurons in group $g$. Each neuron's $S^i(t)$ and $U^i(t)$ represent the output spike

and membrane potential of the $i$-th neuron in group $g$ at time $t$.

$$S^i(t) = \begin{cases} 1, & \text{if } U^i(t) \geq V_{th}^g(t-1) \\ 0, & \text{otherwise} \end{cases}, \quad i \in N_g \tag{8}$$

$$S^g(t) = \frac{1}{|N_g|} \sum_{i \in N_g} S^i(t) \tag{9}$$

$$V_{th}^g(t) = \gamma S^g(t) + (1-\gamma) V_{th}^g(t-1), \quad V_{th}^g(0) = V_{base} \ \forall g. \tag{10}$$

As depicted above, all neurons in group $N_g$ share the same threshold voltage $V_{th}^g(t)$ that is adjusted in each inference timestep. In the first timestep ($t = 0$), they are set to a hyperparameter *inference base threshold* $V_{base}$. For each timestep $t$, the threshold for each group $V_{th}^g(t)$ is updated using the average spikes from previous timestep $S_{th}^g$, adjusted with $\gamma$ which is a hyperparameter denoting adaptive threshold size. Based on the observation ① from Section 3 that the neuron behavior is related to the node degree, this will let neurons in the group collaboratively find an adequate threshold.

The intuition behind the decision is that the spike frequencies of the neurons are closely related to the degree of the associated vertices. By reformulating Equation (7) from a single-neuron perspective, the membrane potential of neuron $i$ can be written as $U_i = \sum_{j \in \text{active}(i)} W_{i,j}$, where active($i$) denotes firing neighbors of $i$. Assuming $W_{i,j} \sim \mathcal{N}(0, \sigma)$, this gives $U_i \sim \mathcal{N}(0, \sigma \cdot |\text{active}(i)|)$. As $|\text{active}(i)|$ scales with the degree of vertex associated with neuron $i$, grouping neurons by degree clusters those with similar membrane potential variance, which aligns with the observations from Section 3.

The major advantage of this scheme is that it is straightforward to put an unseen node or an unseen graph into a group at inference because only the degree information is required to assign a vertex into a group. To further consider intra-node deviation, we split the group into $F$ (number of features) neurons, which is a fixed parameter determined by the model architecture. For any unseen node, finding out its degree is trivial because visiting its neighbors is one of the fundamental requirements of graph data structures (Khorasani et al., 2014; Wang et al., 2016; Lee et al., 2017).

### 4.3  LEARNABLE INFERENCE BASE THRESHOLD

The proposed group-adaptive threshold scheme effectively reduces the spike frequency variation issue. However, we find that the adaptive neurons in the proposed DAG are sensitive to their inference base threshold ($V_{base}$), which is a hyperparameter. As depicted in Figure 3, the performance of the adaptive neurons can severely drop when $V_{base}$ is not carefully tuned, which aligns with the findings from (Bellec et al., 2018). Moreover, manually tuning the inference base thresholds individually for the groups would be difficult because there are thousands of neuron groups.

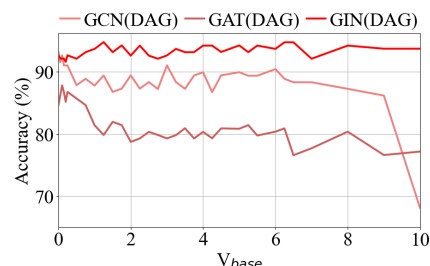

Figure 3: Sensitivity of neurons to its inference base threshold.

To address the problem, we propose *learnable inference base threshold* (LIBT), which makes the inference base thresholds per group ($V_{base}^g$) and the decay rate ($\beta$) learnable. Even though the base thresholds of all groups are initialized to the same value $V_{init}$ at epoch 0, they will be updated differently during training to reflect the characteristics of each neuron group. Considering this, Equation (10) can be rewritten to:

$$V_{th}^g(t) = \gamma S^g(t) + (1-\gamma) V_{th}^g(t-1), \quad V_{th}^g(0) = V_{base}^g. \tag{11}$$

For the inference phase, we use the $V_{base}^g$ values learned during the training phase, which are adjusted for each group. The algorithm for overall training procedure is presented in Appendix C.

## 5  EVALUATION

### 5.1  EXPERIMENTAL SETTINGS

We use a total of five graph datasets commonly used for benchmarking GNNs: MUTAG (Debnath et al., 1991), PROTEINS (Borgwardt et al., 2005), ENZYMES (Borgwardt et al., 2005), NCI1 (Wale

Table 1: Performance comparison against baseline methods.

| Model | Method | MUTAG | PROTEINS | ENZYMES | NCI1 | IMDB-BINARY |
|---|---|---|---|---|---|---|
| GCN | ANN Kipf & Welling (2016) | 90.47 ± 6.60 | 77.81 ± 3.46 | 69.50 ± 5.33 | 78.93 ± 2.98 | 56.80 ± 4.80 |
| | Vanilla LIF - SpikingGNN | 90.96 ± 3.99 | 74.39 ± 2.68 | 55.33 ± 4.31 | 73.41 ± 1.60 | 68.40 ± 2.96 |
| | Adaptive LIF - SpikeNet | 87.81 ± 5.60 | 74.75 ± 3.20 | 56.33 ± 4.08 | 73.92 ± 1.54 | 70.30 ± 2.17 |
| | PLIF - PGNN | 87.28 ± 5.87 | 77.36 ± 2.68 | 60.50 ± 3.60 | 76.52 ± 1.46 | 71.60 ± 2.17 |
| | Gated LIF - GGNN | 89.39 ± 5.21 | 77.00 ± 3.02 | **60.87** ± 3.47 | 75.52 ± 1.77 | 70.10 ± 3.30 |
| | DAS-GNN | **97.37** ± 3.20 (+6.41) | **77.72** ± 2.64 (+0.36) | 60.17 ± 3.20 (-0.70) | **77.25** ± 1.66 (+0.73) | **80.60** ± 2.51 (+9.00) |
| GAT | ANN Veličković et al. (2018) | 91.02 ± 5.62 | 77.54 ± 3.22 | 61.83 ± 4.37 | 73.75 ± 1.21 | 54.80 ± 2.14 |
| | Vanilla LIF - SpikingGNN | 78.71 ± 5.34 | 59.66 ± 0.21 | 27.67 ± 3.65 | 66.25 ± 1.77 | 50.00 ± 0.00 |
| | Adaptive LIF - SpikeNet | 78.22 ± 3.67 | 64.60 ± 3.22 | 58.17 ± 3.39 | 66.84 ± 1.60 | 50.00 ± 0.00 |
| | PLIF - PGNN | 82.49 ± 4.98 | 64.06 ± 2.37 | 38.50 ± 3.79 | 68.32 ± 1.49 | 50.00 ± 0.00 |
| | Gated LIF - GGNN | 83.07 ± 5.16 | 69.89 ± 3.32 | 60.33 ± 3.11 | 67.79 ± 2.01 | 50.00 ± 0.00 |
| | DAS-GNN | **94.21** ± 4.14 (+11.14) | **72.14** ± 2.99 (+2.25) | **61.00** ± 3.38 (+0.67) | **73.82** ± 1.67 (+5.50) | **77.80** ± 1.69 (+27.80) |
| GIN | ANN Xu et al. (2019) | 96.32 ± 3.10 | 78.79 ± 3.74 | 69.17 ± 3.90 | 79.17 ± 3.07 | 73.30 ± 2.80 |
| | Vanilla LIF - SpikingGNN | 92.60 ± 4.41 | 77.81 ± 2.71 | 47.33 ± 3.14 | 70.29 ± 2.01 | 74.30 ± 1.47 |
| | Adaptive LIF - SpikeNet | 93.66 ± 4.62 | 78.43 ± 2.63 | 50.67 ± 3.61 | 74.77 ± 1.63 | 74.80 ± 2.74 |
| | PLIF - PGNN | 94.18 ± 4.84 | 79.16 ± 2.61 | 50.17 ± 3.71 | 75.38 ± 1.41 | 72.80 ± 4.63 |
| | Gated LIF - GGNN | 93.66 ± 3.74 | 79.13 ± 2.28 | 49.50 ± 3.20 | 76.13 ± 1.40 | 76.80 ± 1.25 |
| | DAS-GNN | **96.32** ± 3.46 (+2.13) | **80.02** ± 2.49 (+0.86) | **57.83** ± 3.08 (+7.17) | **77.45** ± 1.30 (+1.31) | **79.40** ± 2.41 (+2.60) |

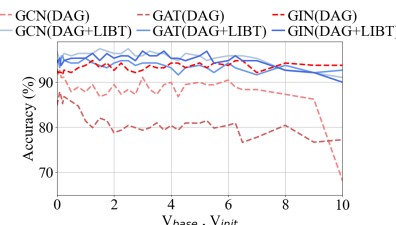

Figure 4: Sensitivity of hyperparameters associated with the threshold voltage value. $V_{base}$ for DAG, $V_{init}$ for DAG+LIBT.

Table 2: Sensitivity analysis of learning rate.

| Model | learning rate ($\eta$) | | | | | |
|---|---|---|---|---|---|---|
| | 0.001 | 0.005 | 0.01 | 0.05 | 0.1 | 0.5 |
| GCN | 87.78 | 97.37 | 97.37 | 96.84 | 95.79 | 83.01 |
| GAT | 85.73 | 93.68 | 94.21 | 94.71 | 95.26 | 90.99 |
| GIN | 92.54 | 95.79 | 96.32 | 92.60 | 92.08 | 89.42 |

Table 3: Sensitivity analysis of adaptive threshold size.

| Dataset | Model | adaptive threshold size ($\gamma$) | | | | | | | |
|---|---|---|---|---|---|---|---|---|---|
| | | 0.05 | 0.10 | 0.15 | 0.20 | 0.25 | 0.30 | 0.35 | 0.40 |
| MUTAG | GCN | 95.79 | 96.84 | 96.32 | 97.37 | 93.13 | 96.32 | 96.29 | 95.76 |
| | GAT | 95.23 | 93.66 | 92.60 | 94.21 | 94.74 | 92.60 | 92.60 | 92.08 |
| | GIN | 96.84 | 95.26 | 95.79 | 96.32 | 95.26 | 95.26 | 95.26 | 95.79 |

et al., 2008), and IMDB-Binary (Yanardag & Vishwanathan, 2015b). We use three different GNN layers: GCN (Kipf & Welling, 2016), GAT (Veličković et al., 2018), and GIN (Xu et al., 2019). We choose four SNN baselines that are applicable to graph datasets: SpikingGNN (Zhu et al., 2022), SpikeNet (Li et al., 2023), PGNN (Fang et al., 2021) and GGNN (Yao et al., 2022). Since this is the first SNN design to target graph classification, we apply minor modifications to each architecture, such as appending a readout layer. Note that SpikingGNN (Zhu et al., 2022) was originally proposed for GCN, but we extend it to both GAT and GIN. For the GCN architectures, we use 3 layers with 128 hidden dimensions. For the GIN architectures, we use 2 layers with 128 hidden dimensions where each layer comprises 2 sublayers of MLP in their combination stages. Lastly, GAT architectures have 2 layers with 4-multi head attentions with 64 dimensions. More details are in Appendix B.

## 5.2 RESULTS ON GRAPH CLASSIFICATION

We compare DAS-GNN against prior works that adopt a spiking neural network to graph datasets, as shown in Table 1. We also report the performance of conventional ANN for comparison. In all but a single case, DAS-GNN outperforms the baselines by a noticeable margin. In the case where DAS-GNN underperforms, the gap is only 0.70%p, smaller than the error bounds. In the opposite cases, the improvement is up to 27.80%p, showing a great amount of improvement.

An intriguing result is that DAS-GNN performs better than ANN-based GNNs in several cases, contrary to usual cases where ANNs serve as an upper performance bound of SNNs. Improvements beyond the error bounds are found in MUTAG (GCN and GAT), NCI1 (GAT), and IMDB-BINARY (GCN, GAT, and GIN). Note that the model architecture and the number of learnable parameters are the same in all methods. We believe this could come from the spiking neurons efficiently capturing the irregular connections over several timesteps, thereby showing an advantage over ANNs.

## 5.3 SENSITIVITY STUDY

In the proposed DAG and LIBT techniques, there are hyperparameters associated with the threshold voltage values. For DAG, $V_{base}$ determines the thresholds at the first inference step, and for LIBT,

Table 4: Ablation study on the proposed method.

| Model | DAG | LIBT | MUTAG | PROTEINS | ENZYMES | NCI1 | IMDB-BINARY |
|-------|-----|------|-------|----------|---------|------|-------------|
| GCN | ✗ | ✗ | 90.96 | 74.39 | 55.33 | 73.41 | 68.40 |
|     | ✓ | ✗ | 93.66 (+2.70) | 75.65 (+1.26) | 56.83 (+1.50) | 73.65 (+0.24) | 71.90 (+3.50) |
|     | ✗ | ✓ | 95.26 (+4.30) | 75.83 (+1.44) | 59.00 (+3.67) | 74.14 (+0.73) | 80.10 (+11.70) |
|     | ✓ | ✓ | 97.37 (+6.41) | 77.72 (+3.33) | 60.17 (+4.84) | 77.25 (+3.84) | 80.60 (+12.20) |
| GAT | ✗ | ✗ | 78.71 | 59.66 | 27.67 | 66.25 | 50.00 |
|     | ✓ | ✗ | 80.35 (+1.64) | 66.48 (+6.82) | 57.83 (+30.16) | 67.98 (+1.73) | 50.00 (+0.00) |
|     | ✗ | ✓ | 93.68 (+14.97) | 64.96 (+5.30) | 52.17 (+24.50) | 69.61 (+3.36) | 75.80 (+25.80) |
|     | ✓ | ✓ | 94.21 (+15.50) | 72.14 (+12.48) | 61.00 (+33.33) | 73.82 (+7.57) | 77.80 (+27.80) |
| GIN | ✗ | ✗ | 92.60 | 77.81 | 47.33 | 70.29 | 74.30 |
|     | ✓ | ✗ | 93.66 (+1.06) | 78.35 (+0.54) | 50.83 (+3.50) | 73.67 (+3.38) | 75.20 (+0.90) |
|     | ✗ | ✓ | 94.71 (+2.11) | 79.33 (+1.52) | 48.50 (+1.17) | 71.14 (+0.85) | 74.00 (-0.30) |
|     | ✓ | ✓ | 96.32 (+3.72) | 80.02 (+2.21) | 57.83 (+10.50) | 77.45 (+7.16) | 79.40 (+5.10) |

$V_{init}$ determines the threshold initialization value at epoch 0. To see if the LIBT addresses the sensitivity issue of DAG mentioned in Section 4.3, we varied those hyperparameters from 0.0 to 10.0 and measured the accuracy. Figure 4 depicts the results of the sensitivity study. As opposed to the DAG, which is very sensitive to $V_{base}$ (dashed lines), DAG +LIBT makes the accuracy stable to $V_{init}$ (solid lines). Moreover, the peak accuracies are significantly higher for DAG +LIBT.

Since our scheme uses a learnable inference base threshold, we also study its sensitivity for the learning rate, shown in Table 2. As shown in the table, DAS-GNN is relatively insensitive to learning rate, except for the extreme values (0.001 or 0.5). As denoted in the experimental setting, we use $\eta = 0.01$ as the default value based on the results. In addition, we conduct sensitivity study of $\gamma$, which adjusts the amount of how much thresholds change in our DAG method, shown in Table 3. We find that DAG is generally insensitive to $\gamma$, with minimal degradation within the given range. We use 0.2 as the default value of $\gamma$, which is generally the best setting across different model architectures.

## 5.4 ABLATION STUDY

In this section, we break down individual components of DAS-GNN and perform an ablation study, which is reported in Table 4. Starting from baseline implementation, which does not differentiate neurons used by each node, we apply DAG to show the effect of degree-aware group-adaptive neurons. Then, we experimented with our LIBT scheme to degree-divided groups without DAG. Lastly, we applied both of our DAG and LIBT to compose DAS-GNN.

The results show that DAG alone can improve the performance across all datasets and models. This means that uneven spike distribution caused by indegree variance is a general problem shared across different graph datasets, and simply grouping the nodes with similar indegree to share the same threshold helps alleviate this problem. In addition, adding a LIBT scheme not only makes it less sensitive for inference base threshold values but also further boosts the accuracy in almost all cases, demonstrating its efficacy and stability.

## 5.5 SPIKE FREQUENCY DISTRIBUTION ANALYSIS

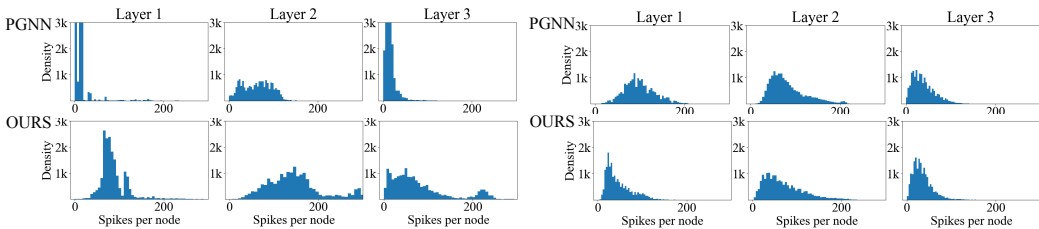

(a) Spike frequency distribution histogram of PGNN and DAS-GNN on the IMDB-BINARY dataset.

(b) Spike frequency distribution histogram of PGNN and DAS-GNN on the ENZYMES dataset.

Figure 5: Spike distribution histogram for PGNN and DAS-GNN (ours)

Table 5: Energy consumption comparison against baseline methods.

| Model | Method | MUTAG | PROTEINS | ENZYMES | NCI1 | IMDB-BINARY |
|-------|--------|-------|----------|---------|------|-------------|
| | ANN (Kipf & Welling, 2016) | 0.53mJ | 6.92mJ | 3.29mJ | 21.41mJ | 3.16mJ |
| GCN | SpikingGNN (Zhu et al., 2022) | 0.02mJ | 1.28mJ | 0.36mJ | 3.92mJ | 0.10mJ |
| | SpikeNet (Li et al., 2023) | 0.06mJ | 0.13mJ | 0.25mJ | 2.64mJ | 0.35mJ |
| | PGNN (Fang et al., 2021) | 0.10mJ | 0.79mJ | 0.84mJ | 6.54mJ | 0.28mJ |
| | GGNN (Yao et al., 2022) | 0.08mJ | 0.91mJ | 0.48mJ | 3.37mJ | 0.21mJ |
| | +Additional cost | 0.02 $\mu$J | 0.08$\mu$J | 0.05$\mu$J | 0.02$\mu$J | 0.30$\mu$J |
| | DAS-GNN | 0.10mJ (-82.16%) | 0.94mJ (-86.36%) | 0.52mJ (-84.29%) | 5.28mJ (-75.35%) | 0.70mJ (-77.71%) |

In this section, we perform additional analyses on DAS-GNN by studying its spike frequency distribution. In Figure 6, we provide the same spike frequency visualization as done in Section 3, but using DAS-GNN and the best-performing baseline PGNN (Fang et al., 2021). Unlike Figure 1, which showed severe starvation with most nodes not generating spikes, Figure 5 reveals that most nodes in DAS-GNN fire a meaningful number of spikes, significantly alleviating the starvation problem. This is further illustrated in Figure 6, where most neurons have non-zero spike values and successfully reflect the topology of the graph. For nodes with higher degrees, the spikes are more frequent (close to 5) due to having more incoming spikes from their neighbors.

In addition, we observed some correlation between the spike diversity, especially in the last layer, and model performance as shown in Figure 5. In the ENZYMES dataset, both DAS-GNN and PGNN show highly diversified spike distributions, which lead to similar performance. However, in the IMDB-BINARY dataset, PGNN fails to achieve the same level of spike diversity, which does not perform well enough compared with DAS-GNN. Our DAS-GNN design effectively integrates node degree information and propagates it through spikes, thus reflecting graphs' topology well.

## 5.6 ENERGY CONSUMPTION ANALYSIS

To determine whether DAS-GNN attains similar energy consumption benefits that are typically seen in SNN, we calculated the theoretical energy consumption as shown in Table 5. These calculations are based on Horowitz (2014); Yao et al. (2023), which are widely used for SNN energy consumption analysis. We calculated energy consumption based on each layer's spike sparsity $\mu$ and FLOPs (floating point operations). Assuming MAC and AC operations are implemented on 45nm hardware, we used $E_{MAC} = 4.6pJ$ and $E_{AC} = 0.9pJ$. The theoretical energy consumption for the SNN was calculated with $E_{AC} \times \mu \times FLOPs$.

In the table, the 'Additional Cost' row represents the overhead caused by updating the per-group threshold as in Equation (11). The additional cost calculated by $2 \times E_{AC} \times$ (number of unique degrees) $\times$ (hidden dimension) based on Equation (10). To share the per-group threshold among neurons without communication, we assume that neurons within a group are mapped to the same processing element. This way, the shared threshold value is stored in the local memory of the processing element of neuromorphic hardwares (Merolla et al., 2014; Davies et al., 2018; Akopyan et al., 2015), and the firing rate can be updated simply by counting.

As shown in Table 5, DAS-GNN achieves superior energy efficiency compared to the ANN counterparts of around 80% improvements in all datasets. In addition, DAS-GNN maintains energy consumption comparable to other baseline SNN models despite diversifying the spike frequency distributions. It is worth noting that the additional cost for adaptively updating the thresholds is almost negligible. This is because the update operations happen only once per group, whereas the base computation requires operations per every connection for individual neurons. Because a group typically has a few thousands of neurons and each neuron has multiple connections, the overhead becomes an order of magnitude smaller compared to that of the main computation. For more energy consumption results, please refer to Table 14 in the Appendix J.

## 6 RELATED WORKS

**Graph Classification** Graph classification requires identifying the global characteristics of each graph and is commonly applied to domains such as bioinformatics (Borgwardt et al., 2005), chemoinformatics (Zhu et al., 2012), or social network analysis (Hamilton et al., 2017; McCal-

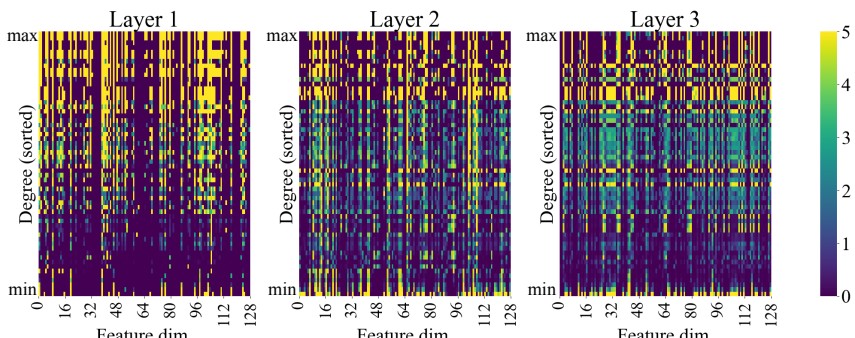

Figure 6: Analysis on spike frequency variation of GCN on IMDB-BINARY dataset using DAS-GNN. X-axis denotes feature dimension, while Y-axis denotes nodes grouped and sorted by degree in descending order, top to bottom. Brighter spots denote higher frequency.

lum et al., 2000). Popular examples include the molecular classification of chemical compounds, proteins, or RNAs, where identifying the graph structural information is crucial. Many GNNs (Kipf & Welling, 2016; Veličković et al., 2018; Xu et al., 2019; Yoo et al., 2023) use a message-passing paradigm (Gilmer et al., 2017) to aggregate local features. Thus, to obtain global features representing the entire graph, graph pooling (Ying et al., 2018) is often used. Global pooling summarizes the entire graph into a fixed-size graph embedding, which can be done by simply averaging or taking minimum or maximum values of the node embeddings. Other variations replace such simple operations with neural networks (Vinyals et al., 2015; Li et al., 2016) or integrate sorting to selectively choose which node embeddings to include Zhang et al. (2018). More advanced techniques such as hierarchical pooling utilize hierarchical information of graphs (Ranjan et al., 2020; Lee et al., 2019; Gao & Ji, 2019; Diehl, 2019) and usually show better representation learning (Zhang et al., 2018).

**Spiking Neural Networks** SNNs are a type of neural network where information is transmitted via spikes, similar to how biological neurons work. They use different neuron models for capturing spike signals effectively (Hodgkin & Huxley, 1952; Hunsberger & Eliasmith, 2015) or adjusting parameters dynamically to complement the accuracy (Fang et al., 2021; Wang et al., 2022; Bellec et al., 2018; Lian et al., 2024). One major research area is converting traditional ANNs into SNNs by mapping ANN activation functions into spike signals (Han et al., 2020; Rueckauer et al., 2017; Hunsberger & Eliasmith, 2015; Sengupta et al., 2019; Fang et al., 2023). Another focus is directly training SNNs using backpropagation, which involves using various techniques such as surrogate functions for backpropagation (Shrestha & Orchard, 2018; Che et al., 2022) and adapting normalization techniques (Sengupta et al., 2019; Duan et al., 2022; Jiang et al., 2024; Zhu et al., 2024).

**SNN for Graphs** Previous attempts to apply SNNs to graph datasets have primarily focused on node-level classification tasks (Zhang et al., 2024; Sun et al., 2024; Zhu et al., 2022; Xu et al., 2021) and have not yet been extended to graph-level tasks. While Wang & Jiang (2022) explored the application of spike training to Graph Attention Networks (GAT), it implemented the message passing phase after the spiking phase, which deviates from previous structures. Additionally, recent efforts have begun to integrate SNNs with other techniques for contrastive learning (Li et al., 2024), particularly in dynamic graphs (Yin et al., 2024), to adopt collaboration between GNNs and SNNs.

## 7 CONCLUSION

In this paper, we explore the application of SNNs to graph neural networks for graph classification for the first time. After thoroughly analyzing the graph's uneven spike distribution, we identify that the degree of each node correlates to this phenomenon. To better accommodate such characteristics of graphs, we propose degree-aware group-adaptive neurons, where we place neurons from vertices sharing the same degree into groups. In addition, we propose to learn the inference base threshold and adaptively adjust the threshold simultaneously to reduce its sensitivity and facilitate training using spikes. Combined with the modified architecture for graph classification, the proposed DAS-GNN outperforms existing works by a noticeable margin.

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

## A  CODE

The code implementing our work is provided in a zip archive as part of the supplementary material. The code is under Nvidia Source Code License-NC and GNU General Public License v3.0. We have made significant efforts to ensure the reproducibility of our work for other researchers. Detailed information on the experimental setup can be found in Appendix B.

## B  DETAILED EXPERIMENT SETTINGS

**Dataset Details**  Given the diverse characteristics of graph datasets, we selected five well-known datasets from TUDatasets, which are commonly used for graph classification. We also compiled statistics for these datasets to briefly summarize their key properties.

Table 6: Summary of datasets used in the study.

| Dataset | # Graphs | Avg. Nodes | # Nodes ($1^{st} graph$) | Avg. Edges | # Edges ($1^{st} graph$) | # Classes |
|---|---|---|---|---|---|---|
| MUTAG Debnath et al. (1991) | 188 | 17.93 | 17 | 19.79 | 38 | 2 |
| PROTEINS Borgwardt et al. (2005) | 1113 | 39.06 | 42 | 72.82 | 162 | 2 |
| ENZYMES Borgwardt et al. (2005) | 600 | 32.6 | 37 | 62.1 | 168 | 6 |
| NCI1 Wale et al. (2008) | 4110 | 29.87 | 21 | 32.30 | 42 | 2 |
| IMDB-BINARY Yanardag & Vishwanathan (2015b) | 1000 | 19.77 | 20 | 96.53 | 146 | 2 |

**Network Architecture**  In this work, we consider the following three GNN architectures where the distinctions lie in their update rules:

- Graph Convolution Network (Kipf & Welling, 2016) (GCN):
  $h_i^{(l+1)} = \sigma(\sum_{j \in \mathcal{N}(i) \bigcup \{i\}} \frac{W h_j^{(l)}}{\sqrt{|N(i)||N(j)|}})$, where $\phi(\cdot)$ is replaced by affine transformation $W$ followed by nonlinearity $\sigma$.

- Graph Attention Network (Veličković et al., 2018) (GAT):
  $h_i^{(l+1)} = \alpha_{i,i} W h_i^{(l)} + \sum_{j \in \mathcal{N}(i)} \alpha_{ij} W h_j^{(l)}$, where $\alpha_{ij}$ is the normalized attention score between node $i$ and $j$.

- Graph Isomorphism Network (Xu et al., 2019) (GIN):
  $h_i^{(l+1)} = MLP((1 + \epsilon)h_i^{(l)} + \sum_{j \in \mathcal{N}(i)} h_j^{(l)})$, where $\epsilon$ is a learnable constant.

For the GCN architectures, we used 128 dimensions for hidden dimension size, and 3 layers were composed of our network. For the GAT architectures, we used 2 layers, 64 dimensions for hidden dimension size with 4 multi-head attentions. For the GIN architectures, our network is composed of 2-MLP layers with a hidden dimension size of 128 for the above equation. We extended Spiking-GCN into GAT and GIN architectures as it was not introduced in the original paper.

The baseline settings, SpikeNet, PGNN, GGNN are the neuron-variant models that replace each SNN layer with different types of neurons. SpikeNet successfully adopted SNN into temporal GNN domains by proposing adaptive LIF neuron models. We adopted adaptive LIF into the static graph datasets since it shows state-of-arts performance in the SGNN area. Another effective neuron model is the parametric leaky integrate-and-fire (PLIF), widely adopted to other SNN tasks such as computer vision and etc. They make the neuron's membrane time constant into a learnable parameter which makes it robust on the initial values. GLIF is another effective neuron model that enhances the idea of PLIF neuron models by adopting linear leakage, introduced to fully parameterized neuron. These three neuron models were selected to compare with our DAS-GNN which showed competitive performance for the neuron-variant models. For all experiments including both baselines and ours, we kept the GNN architecture unchanged (GCN, GAT, GIN), and only replaced the neurons with the ones proposed in each method.

**Experimental Settings**  We trained and evaluated our models using 10-fold cross-validation across all datasets. For the IMDB-BINARY dataset, which lacks inherent features, we generated features to 1 to use node degrees for the GNN layer. Additionally, we did not apply any multiplier to adjust

the width of the sigmoid function. The details of our evaluation procedure are outlined below. All experiments were conducted on a single RTX 4090 GPU for full-batch GNN training.

- Epochs: 1000
- Surrogate function: $\sigma(x) = \frac{1}{1+e^{-x}}$
- Learning rate ($\eta$): 0.01 (for main table)
- Optimizer: Adamw
- Loss function: Cross-entropy
- Adaptive threshold size ($\gamma$): 0.2

## C OVERALL TRAINING PROCEDURE

As referred to in Section 4 our DAG method and overall updating initial values of group threshold are referred to in Algorithm 1. Note that our initial group values are updated after timestep T.

---

**Algorithm 1** Updataing inference base threshold $V_{th}^g(0)$ procedure

---

1: **Inputs:** Initial start points of training threshold $V_{init}$, graph's vertex feature $X \in R^{NXF}$, learning rate for training $\eta$, total time step $T$, $l$-th layer's threshold $V_{th}^{(l)}$, $l$-th layer's message passing operation $MSG^{(l)}$, true label $Y$,
2: **Initialize:** $V_{th}^g(0) = [V_{init}, ... V_{init}]$          ▷ Initialize all of the g threshold groups with initial values
3: **for** $ep = 1$ to $epochs$ **do**
4:     **for** $t = 1$ to $T$ **do**
5:         $X = PoissonEncoder(X)$          ▷ Generate spikes for the input layer with Poisson encoder
6:         **for** $l = 1$ to $L$ **do**
7:             **for** $g$ in group $G$ **do**
8:                 $X^{g,(l)} = MSG^{(l)}(X^{g,(l)})$                    ▷ Operate by GCN, GAT, GIN architectures
9:                 **for** $i = 1$ to $|N_g|$ **do**
10:                     $X^{g_i,(l)} = S^{g_i,(l)}(t) = SN^{(l)}(X^{g_i,(l)})$              ▷ $X^{g_i,(l)}$ represents $i$-th row of $X^{g,(l)}$
11:                     $S^{g,(l)}(t) = \frac{1}{|N_g|}\sum_{i \in N_g} S^{g_i}(t)$
12:                 **end for**
13:                 $V_{th}^{g,(l)}(t) = \gamma V_{th}^{g,(l)}(t-1) + (1-\gamma)S^{g,(l)}(t)$ ▷ Update threshold through DAG Equation (10)
14:             **end for**
15:         **end for**
16:         $O^t \leftarrow FC(POOL(GNN(X^{(L)}))) + O^{t-1}$
17:     **end for**
18:     $V_{th}(0) = V_{th}(0) - \eta\nabla_{V_{th}(0)}\mathcal{L}(O^{t=1}, Y)$
19: **end for**

---

## D ADDITIONAL RESULTS ON THE LARGE SCALE DATASETS

We show additional experiment results on two large graph datasets used for graph classification: REDDIT-BINARY and COLLAB (Yanardag & Vishwanathan, 2015a), in Table 7. We could find that our DAS-GNN still outperforms other baselines in large graphs with skewed degree distribution. In addition, we also run large graph datasets (ogbn-arxiv, ogbn-mag (Hu et al., 2020)) on node classification tasks, where the results further show that DAS-GNN maintains its competitive performance. DAS-GNN still performs the best, but the gap is smaller compared to the graph classification tasks. We believe this is because the node classification depends more on per-node properties rather than the graph structures, benefitting less from the proposed techniques.

## E SENSITIVITY STUDY ON NUMBER OF GROUPS

When using the proposed degree-aware group-adaptive neurons, we create an individual group for each unique degree (per each feature position). While this works well in general, there might be

Table 7: Performance comparison against baseline methods in large scale graphs

| Model | Method | REDDIT-BINARY | COLLAB | OGBN-ARXIV | OGBN-MAG |
|-------|--------|---------------|--------|------------|----------|
| GCN | ANN Kipf & Welling (2016) | 90.45 | 80.60 | 71.55 | 35.10 |
| | Vanilla LIF - SpikingGNN | 81.10 | 67.24 | 53.52 | 27.78 |
| | Adaptive LIF - SpikeNet | 82.60 | 68.78 | 60.18 | 28.82 |
| | PLIF - PGNN | 82.35 | 68.46 | 58.45 | 27.52 |
| | Gated LIF - GGNN | 83.10 | 67.92 | 58.23 | 28.75 |
| | DAS-GNN | **83.50** (+0.40) | **83.52** (+14.74) | **60.84** (+0.66) | **29.73** (+0.91) |
| GAT | ANN Veličković et al. (2018) | 76.30 | 55.60 | 71.06 | 29.60 |
| | Vanilla LIF - SpikingGNN | 50.00 | 52.00 | 52.21 | 14.05 |
| | Adaptive LIF - SpikeNet | 50.05 | 52.00 | 53.67 | 17.30 |
| | PLIF - PGNN | 52.05 | 52.04 | 53.77 | 16.97 |
| | Gated LIF - GGNN | 53.05 | 52.00 | 53.26 | 16.54 |
| | DAS-GNN | **76.25** (+23.20) | **75.58** (+23.54) | **53.94** (+0.17) | **17.37** (+0.07) |
| GIN | ANN Xu et al. (2019) | 80.75 | 75.11 | 61.01 | 27.55 |
| | Vanilla LIF - SpikingGNN | 79.75 | 52.04 | 56.72 | 23.31 |
| | Adaptive LIF - SpikeNet | 79.70 | 53.66 | 55.79 | 23.37 |
| | PLIF - PGNN | 83.30 | 53.02 | 50.88 | 21.64 |
| | Gated LIF - GGNN | 83.70 | 58.94 | 54.88 | 22.39 |
| | DAS-GNN | **84.20** (+0.50) | **70.38** (+11.44) | **59.91** (+3.19) | **23.98** (+0.61) |

situations where we want fewer number of such groups. For such scenarios, we conducted experiments where we limited the number of such groups. When the desired number of groups is lower than the number of unique degrees (i.e., maximum degree), we merge the groups such that each merged group covers an equal range of degrees.

Table 8 shows the results. As expected, using more groups is beneficial for performance in general. Although the best point varies a little, the best performance for each dataset and architecture is usually achieved when the near-maximum number of groups is used. One exception is PROTEINS dataset on GCN, but the differences are small in this case.

Table 8: Comparison on using different number of groups

| Dataset | #Groups | GCN | GAT | GIN |
|---------|---------|-----|-----|-----|
| MUTAG | 1 | 87.81 | 80.88 | 94.71 |
| | 2 | 96.84 | 87.78 | 96.32 |
| | 3 | 93.10 | 95.79 | 95.79 |
| | 4(max) | 97.37 | 94.21 | 96.32 |
| PROTEINS | 1 | 78.89 | 64.33 | 78.89 |
| | 2 | 78.98 | 63.88 | 78.98 |
| | 5 | 75.83 | 67.39 | 75.83 |
| | 10 | 77.45 | 69.55 | 77.45 |
| | 15 | 77.99 | 70.54 | 77.99 |
| | 17(max) | 77.72 | 72.14 | 80.02 |
| ENZYMES | 1 | 58.33 | 41.33 | 45.17 |
| | 2 | 56.50 | 40.50 | 44.33 |
| | 5 | 52.00 | 45.00 | 41.50 |
| | 10(max) | 60.17 | 61.00 | 57.83 |
| NCI1 | 1 | 75.74 | 67.86 | 73.82 |
| | 2 | 75.77 | 68.08 | 75.06 |
| | 3 | 77.86 | 72.48 | 76.86 |
| | 4 | 77.81 | 74.26 | 76.74 |
| | 5(max) | 77.25 | 73.82 | 77.45 |
| IMDB-BINARY | 1 | 71.70 | 50.00 | 74.60 |
| | 2 | 70.40 | 50.30 | 72.90 |
| | 5 | 69.30 | 56.80 | 71.00 |
| | 10 | 66.70 | 56.40 | 66.70 |
| | 20 | 64.00 | 61.30 | 66.20 |
| | 50 | 65.99 | 64.51 | 65.55 |
| | 65(max) | 80.60 | 77.80 | 79.40 |

# F SENSITIVITY STUDY ON HYPERPARAMETERS ASSOCIATED WITH THE THRESHOLD VOLTAGE VALUE

In Figure 4, our experiments were conducted using various inference base thresholds to demonstrate the sensitivity of our DAG and LIBT schemes. Table 9 provides the numerical values used to generate Figure 4. We observed that the DAG GIN architecture shows high sensitivity to changes in the $V_{base}$ 19.8%p change around the threshold ($V_{base}$ for DAG).

Table 9: Sensitivity study of hyperparameters associated with the threshold voltage value.

| Model | Method | $V_{init}$ | | | | | |
|---|---|---|---|---|---|---|---|
| | | 0.50 | 1.50 | 2.50 | 5.00 | 7.00 | 10.00 |
| GCN | DAG | 87.84 | 86.75 | 88.33 | 89.91 | 88.30 | 68.16 |
| | Ours | 95.79 | 97.37 | 97.37 | 95.79 | 95.23 | 90.99 |
| GAT | DAG | 85.70 | 81.96 | 80.35 | 80.85 | 77.72 | 77.19 |
| | Ours | 94.18 | 93.65 | 94.21 | 93.68 | 91.58 | 92.60 |
| GIN | DAG | 92.08 | 93.13 | 92.57 | 94.21 | 92.08 | 93.68 |
| | Ours | 94.18 | 94.74 | 96.32 | 93.68 | 94.71 | 89.94 |

# G SENSITIVITY STUDY ON LEARNING RATE

Our experiments were conducted under various learning rate conditions $\eta \in [0.001, 0.5]$ to assess their impact. As reported in Table 2 for the MUTAG dataset, we also present results for the PROTEINS, ENZYMES, NCI1, and IMDB-BINARY datasets across GCN, GAT, and GIN architectures. Our model's ability to learn $V_{init}$ demonstrates a sensitivity to learning rate similar to other ANN models. We found that the optimal performance was achieved at a learning rate of $\eta = 0.01$.

# H SENSITIVITY STUDY ON ADAPTIVE THRESHOLD SIZE $\gamma$

We perform a sensitivity study by varying $\gamma$ within the range [0.05, 0.40] to evaluate its impact. As reported in Table 11 for the MUTAG dataset, we present additional results for the PROTEINS, ENZYMES, NCI1, and IMDB-BINARY datasets using GCN, GAT, and GIN architectures. The total results are reported in Table 11.

# I SENSITIVITY STUDY ON HIDDEN DIMENSION SIZE

We perform a sensitivity study with varying hidden dimension between [64,256] to study its impact on the GNN performance. The results are presented in Table 12, where we could observe that DAS-GNN maintains its performance across varying hidden dimension sizes. Further, we compare this result with PLIF sensitivity study (Table 13), where ours consistently outperform PLIF neurons across different hidden dimension sizes.

# J ENERGY CONSUMPTION FOR OTHER ARCHITECTURES

As mentioned in Section 5.6, we calculated DAS-GNN energy consumption based on each layer's spike sparsity $\mu$ and FLOPs (floating point operations). Assuming MAC and AC operations are implemented on 45nm hardware, we used $E_{MAC}$ = 4.6pJ and $E_{AC}$ = 0.9pJ. The theoretical energy consumption for the SNN was calculated with $E_{AC} \times \mu \times$ FLOPs.

Our DAS-GNN proposed an additional operation related to adaptively adjusting the threshold. The additional cost for the adaptive threshold operation would be $2 \times E_{AC} \times$ (number of unique degrees) $\times$ (hidden dimension). However, it does not take a lot of portion, as the number of unique degrees is very small compared to the number of vertices and edges.

Table 10: Extended sensitivity study on learning rate.

| Dataset | Model | learning rate ($\eta$) | | | | | |
|---|---|---|---|---|---|---|---|
| | | 0.001 | 0.005 | 0.01 | 0.05 | 0.1 | 0.5 |
| MUTAG | GCN | 87.78 | 97.37 | 97.37 | 96.84 | 95.79 | 83.01 |
| | GAT | 85.73 | 93.68 | 94.21 | 94.71 | 95.26 | 90.99 |
| | GIN | 92.54 | 95.79 | 96.32 | 92.60 | 92.08 | 89.42 |
| PROTEINS | GCN | 75.11 | 76.82 | 77.72 | 77.36 | 76.82 | 65.67 |
| | GAT | 64.14 | 70.35 | 72.14 | 73.23 | 74.93 | 70.53 |
| | GIN | 77.72 | 79.07 | 80.02 | 78.17 | 76.55 | 75.65 |
| ENZYMES | GCN | 45.00 | 51.17 | 60.17 | 56.83 | 54.67 | 29.17 |
| | GAT | 32.00 | 45.00 | 61.00 | 55.83 | 42.67 | 34.33 |
| | GIN | 37.33 | 44.33 | 57.83 | 35.17 | 31.33 | 29.33 |
| NCI1 | GCN | 73.87 | 77.37 | 77.25 | 80.07 | 78.81 | 66.95 |
| | GAT | 66.93 | 73.31 | 73.82 | 76.06 | 73.48 | 66.69 |
| | GIN | 72.80 | 76.57 | 77.45 | 70.54 | 69.05 | 64.94 |
| IMDB-Binary | GCN | 78.90 | 79.90 | 80.60 | 80.50 | 80.60 | 73.60 |
| | GAT | 74.80 | 75.80 | 77.80 | 75.60 | 75.90 | 75.30 |
| | GIN | 74.10 | 73.00 | 79.40 | 75.40 | 74.70 | 73.60 |

Table 11: Sensitivity study on adaptive threshold size for DAS-GNN.

| Dataset | Model | adaptive threshold size ($\gamma$) | | | | | | | |
|---|---|---|---|---|---|---|---|---|---|
| | | 0.05 | 0.10 | 0.15 | 0.20 | 0.25 | 0.30 | 0.35 | 0.40 |
| MUTAG | GCN | 95.79 | 96.84 | 96.32 | 97.37 | 93.13 | 96.32 | 96.29 | 95.76 |
| | GAT | 95.23 | 93.66 | 92.60 | 94.21 | 94.74 | 92.60 | 92.60 | 92.08 |
| | GIN | 96.84 | 95.26 | 95.79 | 96.32 | 95.26 | 95.26 | 95.26 | 95.79 |
| PROTEINS | GCN | 76.64 | 77.09 | 77.36 | 77.72 | 76.91 | 77.18 | 77.36 | 77.09 |
| | GAT | 70.62 | 70.53 | 71.70 | 72.14 | 71.70 | 74.40 | 72.60 | 71.34 |
| | GIN | 79.51 | 79.78 | 79.52 | 80.02 | 79.78 | 79.15 | 79.42 | 78.98 |
| ENZYMES | GCN | 63.17 | 61.50 | 61.33 | 60.17 | 61.67 | 60.83 | 61.17 | 62.33 |
| | GAT | 60.67 | 61.67 | 61.33 | 61.00 | 62.83 | 60.83 | 62.83 | 63.33 |
| | GIN | 55.50 | 54.50 | 54.00 | 57.83 | 56.50 | 57.00 | 51.83 | 50.83 |
| NCI1 | GCN | 76.91 | 76.62 | 76.59 | 77.25 | 76.52 | 76.50 | 75.01 | 74.60 |
| | GAT | 74.70 | 74.96 | 74.14 | 73.82 | 74.31 | 73.82 | 73.09 | 73.33 |
| | GIN | 78.61 | 77.47 | 77.25 | 77.45 | 75.64 | 75.13 | 75.04 | 73.41 |
| IMDB-Binary | GCN | 80.70 | 80.10 | 80.40 | 80.60 | 80.40 | 80.80 | 81.00 | 80.50 |
| | GAT | 75.40 | 75.80 | 76.60 | 77.80 | 78.20 | 77.10 | 77.10 | 78.10 |
| | GIN | 76.70 | 78.00 | 77.70 | 79.40 | 78.70 | 79.00 | 78.40 | 78.10 |

In Table 5, we reported only the results for the GCN architectures in the main text. Additionally, for Table 14, we provide the energy consumption data for other architectures, such as GAT and GIN. Note that all values are measured in mJ, and the values in parentheses represent the energy reduction ratio compared to the ANN baselines.

Table 12: Sensitivity study on hidden dimension size for DAS-GNN.

| Dataset | Model | hidden dimension size | | | |
|---|---|---|---|---|---|
| | | 32 | 64 | 128 | 256 |
| MUTAG | GCN | 94.74 | 95.79 | 97.37 | 96.84 |
| | GAT | 94.74 | 94.21 | 93.68 | 93.68 |
| | GIN | 95.79 | 95.76 | 96.32 | 93.65 |
| PROTEINS | GCN | 76.64 | 77.99 | 77.72 | 77.99 |
| | GAT | 73.58 | 72.14 | 69.91 | 70.23 |
| | GIN | 80.14 | 79.33 | 80.02 | 79.70 |
| ENZYMES | GCN | 53.33 | 56.50 | 60.17 | 64.00 |
| | GAT | 58.33 | 61.00 | 54.83 | 55.83 |
| | GIN | 53.67 | 57.33 | 57.83 | 57.33 |
| NCI1 | GCN | 72.19 | 74.09 | 77.25 | 78.13 |
| | GAT | 75.88 | 73.82 | 75.16 | 75.72 |
| | GIN | 76.81 | 76.11 | 77.45 | 75.67 |
| IMDB-Binary | GCN | 80.30 | 80.30 | 80.60 | 80.90 |
| | GAT | 77.00 | 77.80 | 76.70 | 77.30 |
| | GIN | 80.00 | 78.90 | 79.40 | 76.20 |

Table 13: Sensitivity study on hidden dimension size for PLIF.

| Dataset | Model | Hidden dimension size | | | |
|---|---|---|---|---|---|
| | | 32 | 64 | 128 | 256 |
| MUTAG | GCN | 87.81 | 86.75 | 87.28 | 88.33 |
| | GAT | 83.01 | 82.49 | 83.01 | 83.01 |
| | GIN | 93.13 | 93.13 | 94.18 | 94.18 |
| PROTEINS | GCN | 75.11 | 76.82 | 77.72 | 76.46 |
| | GAT | 68.73 | 64.06 | 67.29 | 66.93 |
| | GIN | 79.07 | 78.44 | 79.16 | 77.81 |
| ENZYMES | GCN | 53.00 | 57.17 | 60.50 | 59.83 |
| | GAT | 37.17 | 38.50 | 37.83 | 36.67 |
| | GIN | 51.83 | 51.50 | 50.17 | 48.67 |
| NCI1 | GCN | 70.39 | 72.60 | 76.52 | 77.69 |
| | GAT | 60.40 | 68.32 | 60.42 | 58.52 |
| | GIN | 73.70 | 73.33 | 75.38 | 72.02 |
| IMDB-Binary | GCN | 66.80 | 68.80 | 71.60 | 71.30 |
| | GAT | 50.00 | 50.00 | 50.00 | 50.00 |
| | GIN | 74.90 | 76.30 | 72.80 | 69.80 |

# K    ADOPTING DIFFERENT SPIKING GRAPH NEURAL NETWORK TECHNIQUES

To show the effectiveness of DAS-GNN, we conducted experiments using three techniques from (Li et al., 2024; Xu et al., 2021) (SpikeGCL, GC-SNN, GA-SNN), with and without our proposed method in Table 15. The results show that DAS-GNN is orthogonal to these techniques and can be additionally applied to further enhance the performance in all the tested cases.

Another dimension of technique we can explore is the temporal encoding of spikes. Although our work focuses on the spike rate, we believe it could be adapted to temporal encoding schemes such as Xiao et al. (2024), by integrating degrees to synaptic delays or encoding community information as a temporal property. These opportunities are orthogonal to this work, and we leave them as our future work.

Table 14: Energy consumption comparison against baseline methods for all architectures.

| Model | Method | MUTAG | PROTEINS | ENZYMES | NCI1 | IMDB-BINARY |
|---|---|---|---|---|---|---|
| | ANN (Kipf & Welling, 2016) | 0.53mJ | 6.92mJ | 3.29mJ | 21.41mJ | 3.16mJ |
| GCN | SpikingGNN (Zhu et al., 2022) | 0.02mJ | 1.28mJ | 0.36mJ | 3.92mJ | 0.10mJ |
| | SpikeNet (Li et al., 2023) | 0.06mJ | 0.13mJ | 0.25mJ | 2.64mJ | 0.35mJ |
| | PGNN (Fang et al., 2021) | 0.10mJ | 0.79mJ | 0.84mJ | 6.54mJ | 0.28mJ |
| | GGNN Yao et al. (2022) | 0.08mJ | 0.91mJ | 0.48mJ | 3.37mJ | 0.21mJ |
| | DAS-GNN | 0.10mJ (-82.16%) | 0.94mJ (-86.36%) | 0.52mJ (-84.29%) | 5.28mJ (-75.35%) | 0.70mJ (-77.71%) |
| | ANN Veličković et al. (2018) | 0.33mJ | 4.59mJ | 2.42mJ | 15.55mJ | 2.20mJ |
| GAT | SpikingGNN Zhu et al. (2022) | 0.01mJ | 0.00mJ | 0.11mJ | 2.26mJ | 0.00mJ |
| | SpikeNet Li et al. (2023) | 0.00mJ | 0.09mJ | 0.28mJ | 5.01mJ | 0.00mJ |
| | PGNN Fang et al. (2021) | 0.09mJ | 0.67mJ | 0.21mJ | 5.04mJ | 0.00mJ |
| | GGNN Yao et al. (2022) | 0.01mJ | 0.85mJ | 0.41mJ | 2.78mJ | 0.05mJ |
| | DAS-GNN | 0.07mJ (-79.96%) | 0.05mJ (-98.82%) | 0.34mJ (-85.83%) | 4.75mJ (-69.44%) | 0.55mJ (-74.89%) |
| | ANN Xu et al. (2019) | 0.39mJ | 4.96mJ | 2.33mJ | 15.26mJ | 2.24mJ |
| GIN | SpikingGNN Zhu et al. (2022) | 0.05mJ | 0.06mJ | 0.19mJ | 0.47mJ | 0.07mJ |
| | SpikeNet Li et al. (2023) | 0.01mJ | 0.14mJ | 0.13mJ | 0.95mJ | 0.07mJ |
| | PGNN Fang et al. (2021) | 0.04mJ | 0.08mJ | 0.17mJ | 1.23mJ | 0.12mJ |
| | GGNN Yao et al. (2022) | 0.03mJ | 0.09mJ | 0.22mJ | 2.51mJ | 0.06mJ |
| | DAS-GNN | 0.05mJ (-87.14%) | 0.02mJ (-99.64%) | 0.14mJ (-96.14%) | 1.67mJ (-89.04%) | 0.06mJ (-97.48%) |

| Model | MUTAG | | PROTEINS | | ENZYMES | | NCI1 | | IMDB-BINARY | |
|---|---|---|---|---|---|---|---|---|---|---|
| | Orig. | DAS-GNN | Orig. | DAS-GNN | Orig. | DAS-GNN | Orig. | DAS-GNN | Orig. | DAS-GNN |
| SpikeGCL | 91.49 | **97.34** | 77.87 | **79.15** | 28.17 | **32.17** | 65.69 | **67.76** | 71.70 | **73.30** |
| GC-SNN | 88.33 | **93.13** | 72.33 | **73.32** | 43.50 | **53.00** | 63.36 | **65.38** | 69.90 | **77.20** |
| GA-SNN | 66.49 | **92.05** | 59.57 | **65.85** | 33.17 | **51.83** | 52.21 | **66.13** | 50.00 | **78.60** |

Table 15: DAS-GNN results across different datasets and models.

## L ANALYSIS ON SPIKE FREQUENCY

We provide additional figures referenced in Section 3. Figure 7 to Figure 28 presents the total spike distribution histogram for the MUTAG, PROTEINS, ENZYMES, NCI1 and IMDB-BINARY datasets. Moreover, we include baselines such as SpikingGNN, SpikeNet, and PGNN to help illustrate the tendencies in spike distribution.

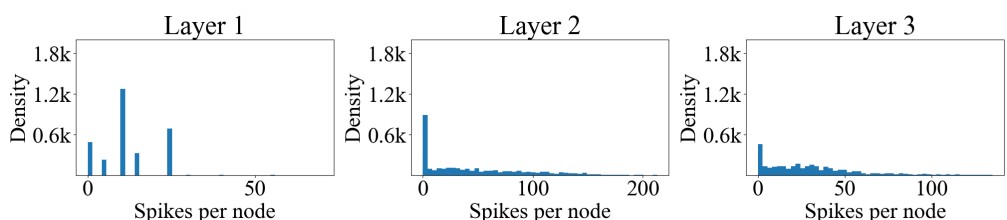

Figure 7: MUTAG-SpikingGNN spike frequency distribution histogram.

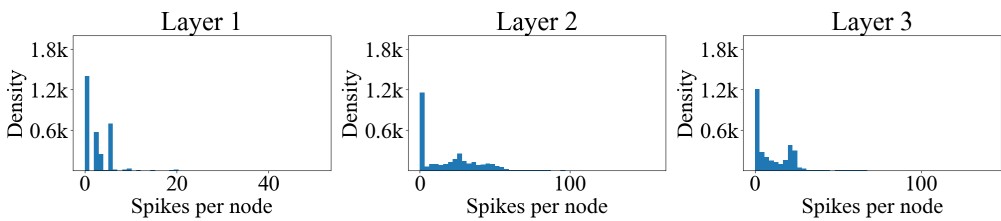

Figure 8: MUTAG-SpikeNet spike frequency distribution histogram.

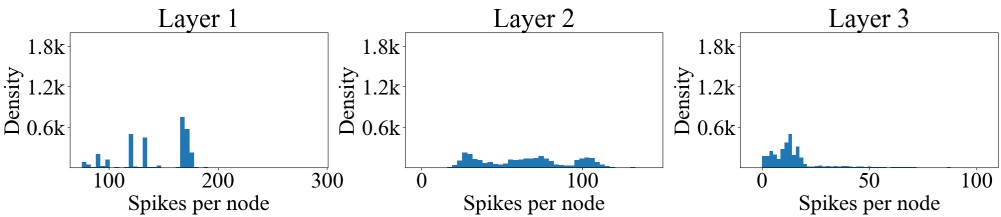

Figure 9: MUTAG-PGNN spike frequency distribution histogram.

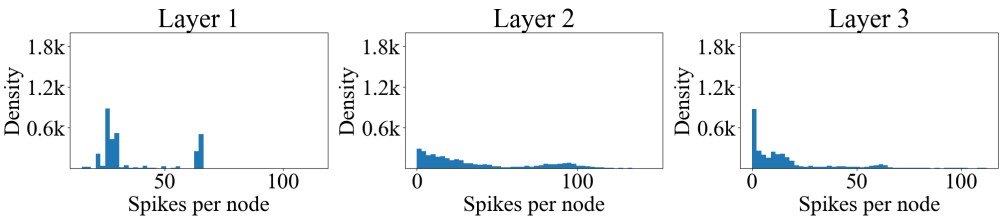

Figure 10: MUTAG-DASGNN spike frequency distribution histogram.

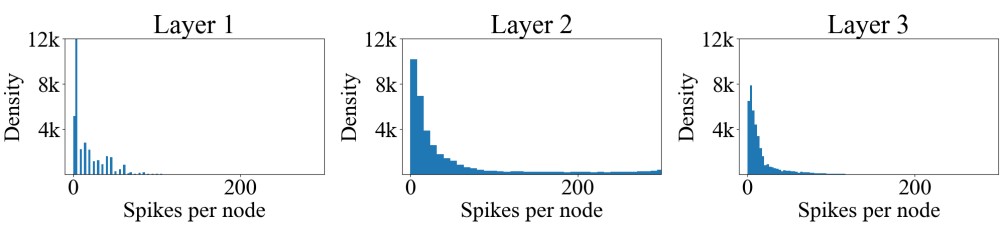

Figure 11: PROTEINS-SpikingGNN spike frequency distribution histogram.

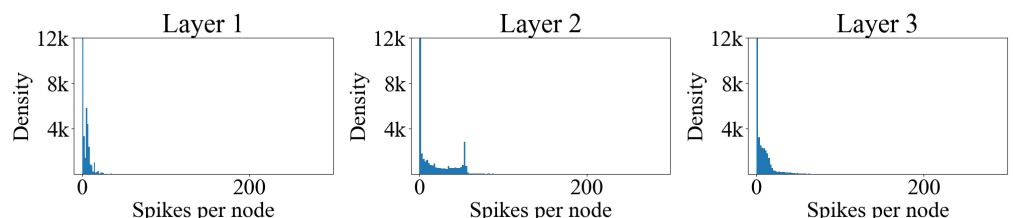

Figure 12: PROTEINS-SpikeNet spike frequency distribution histogram.

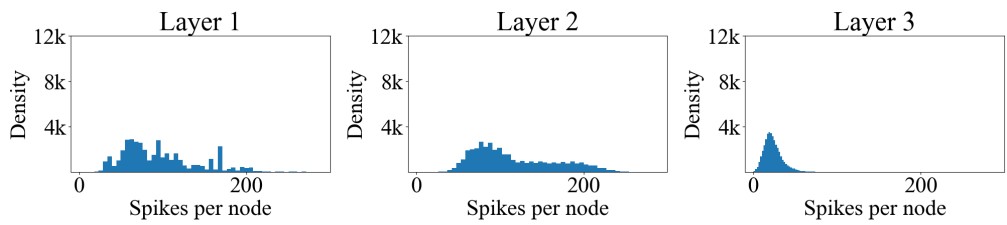

Figure 13: PROTEINS-PGNN spike frequency distribution histogram.

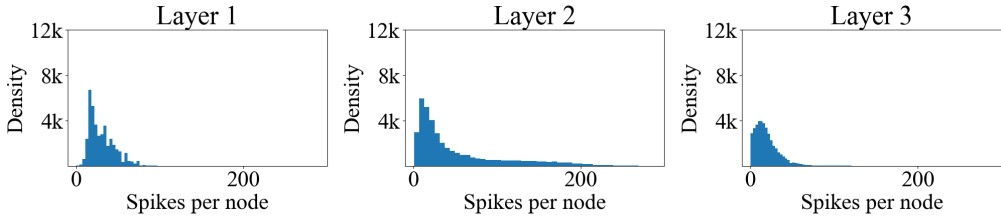

Figure 14: PROTEINS-DASGNN spike frequency distribution histogram.

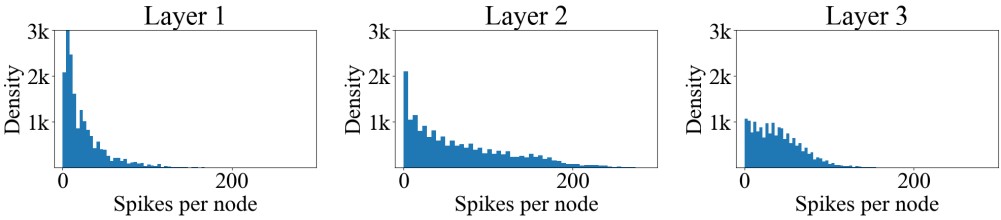

Figure 15: ENZYMES-SpikingGNN spike frequency distribution histogram.

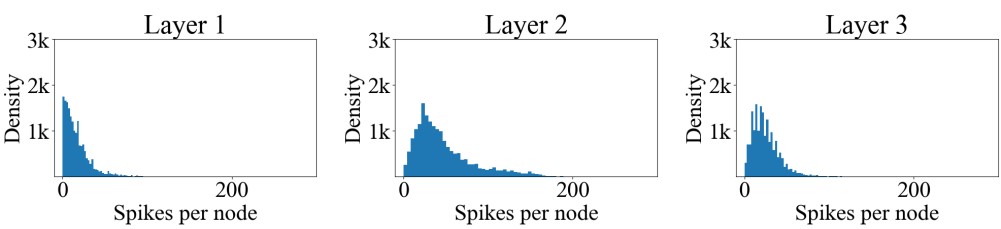

Figure 16: ENZYMES-SpikeNet spike frequency distribution histogram.

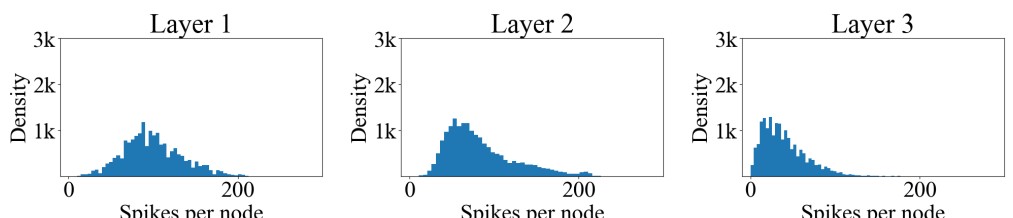

Figure 17: ENZYMES-PGNN spike frequency distribution histogram.

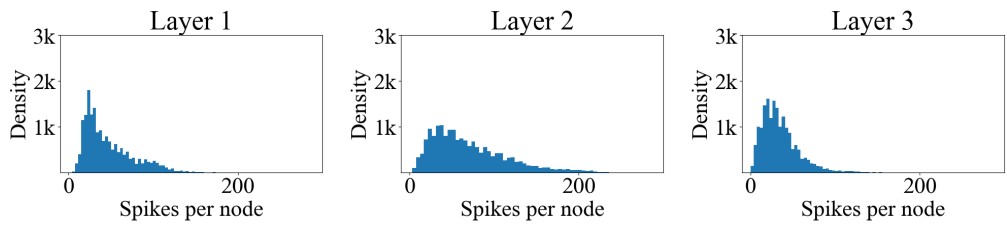

Figure 18: ENZYEMS-DASGNN spike frequency distribution histogram.

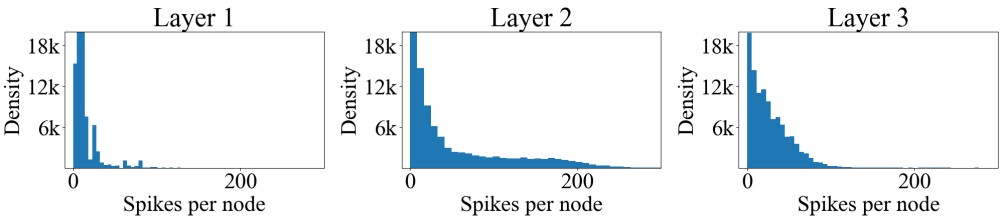

Figure 19: NCI1-SpikingGNN spike frequency distribution histogram.

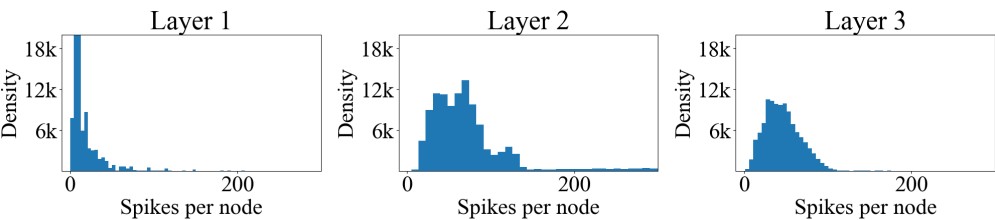

Figure 20: NCI1-SpikeNet spike frequency distribution histogram.

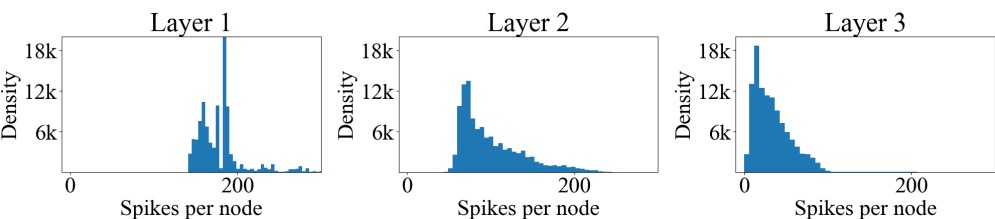

Figure 21: NCI1-PGNN spike frequency distribution histogram.

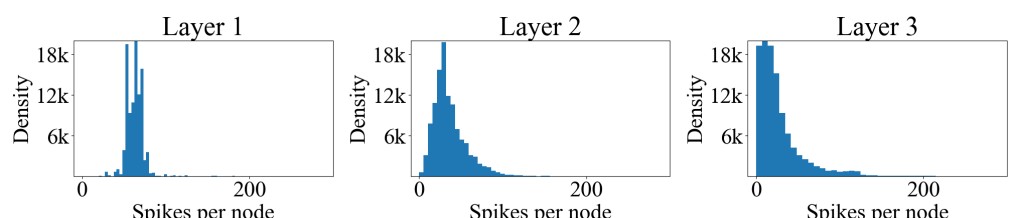

Figure 22: NCI1-DASGNN spike frequency distribution histogram.

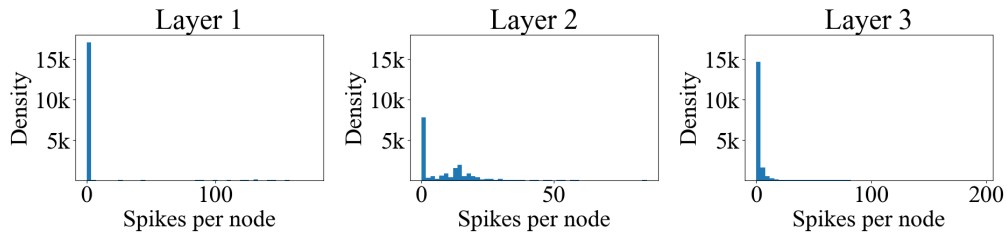

Figure 23: IMDB-BINARY-SpikingGNN spike frequency distribution histogram.

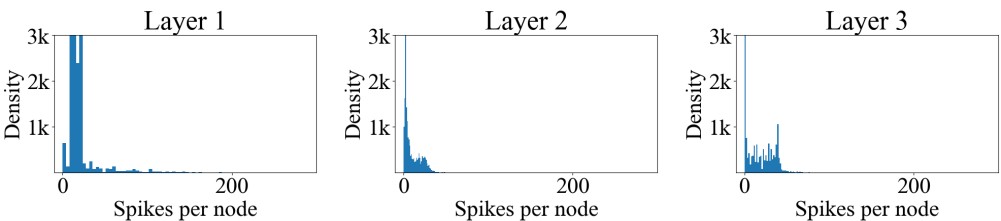

Figure 24: IMDB-BINARY-SpikeNet spike frequency distribution histogram.

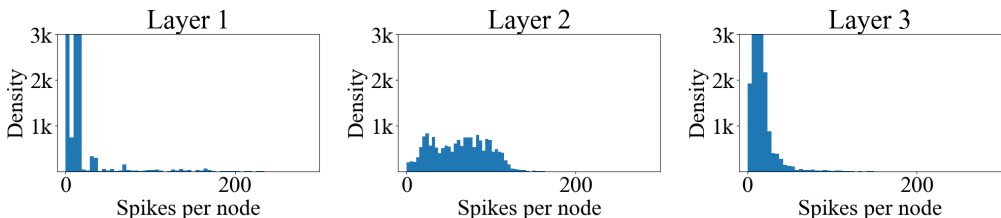

Figure 25: IMDB-BINARY-PGNN spike frequency distribution histogram.

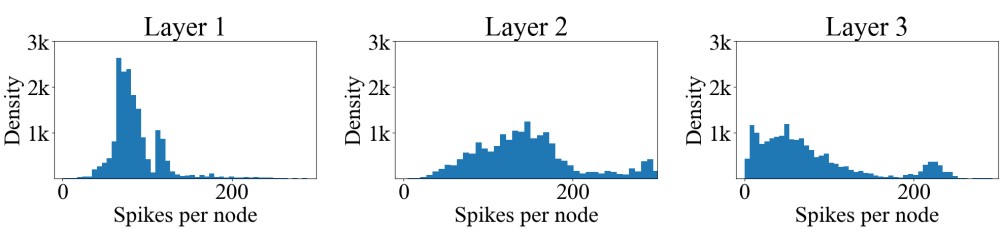

Figure 26: IMDB-BINARY-DASGNN spike frequency distribution histogram.

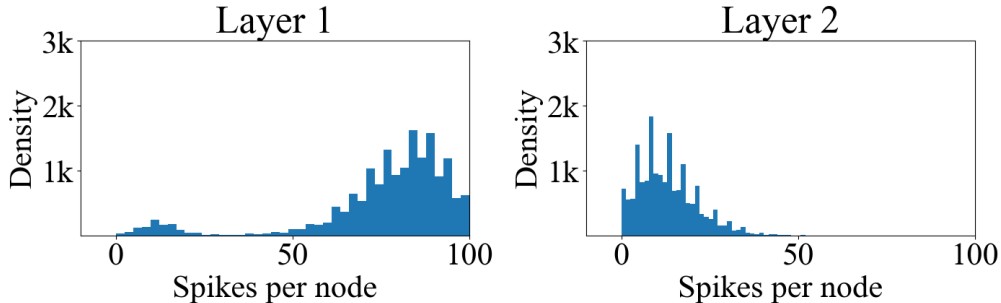

Figure 27: ENZYMES-PGNN-GAT spike frequency distribution histogram.

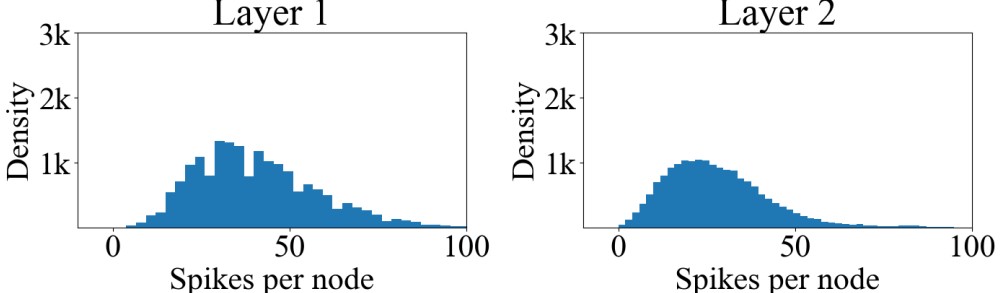

Figure 28: ENZYMES-DASGNN-GAT spike frequency distribution histogram.

