# OpenReview forum: "DAS-GNN: Degree-Aware Spiking Graph Neural Networks for Graph Classification"
_ICLR.cc/2025/Conference — Submitted to ICLR 2025_

### Official Review · Reviewer_skyv · 2024-10-30

**Soundness:** 2
**Presentation:** 2
**Contribution:** 2
**Rating:** 5
**Confidence:** 4

**Summary:**

This paper proposes the starvation problem of spiking neurons, which is widespread in existing spiking graph neural networks. And the paper observes that the spike frequency of neurons is related to the connectivity of the graph and entries in the node feature. Based on these observations, a novel spiking graph representation learning method called DAS-GNN is proposed. DAS-GNN is composed of two modules, degree-aware group adaptive neurons and the learnable inference base threshold. These components are responsible for regulating the threshold voltages of neurons by groups and reducing the influence of the base threshold voltages, respectively. Extensive experiments demonstrate that DAS-GNN outperforms its Artificial Neural Network (ANN) counterparts across some graph classification datasets. These findings highlight the potential of DAS-GNN for energy-saving and accurate graph-based systems.

**Strengths:**

1. This study proposes interesting observations that the connectivity of the graph will affect the receiving and firing patterns of spiking neurons.

2. The authors carefully design two components, degree-aware group adaptive neurons and the learnable inference base threshold, to effectively balance the predictive performance and the energy consumption.

3. Extensive experimental evaluations are provided, which are informative.

**Weaknesses:**

1. Some descriptions of neurons and nodes are unclear. In Section 3, neurons are related to the feature dimension. But, in Section 4.2, the paper claims that neurons can be grouped by their degrees.

2. The definition of the "degree group" is vague and the paper lacks more explanations about the groups $N_g$ or grouping strategies.

3. One of my main concerns is the technical quality. In the experiment, the baseline, PGNN, is always considered as a spiking neuron variant, referred to as Parametric Leaky Integrate-and-Fire model (PLIF) in the previous study [1]. However, the other baselines are spiking graph neural networks, which can adopt PLIF models as their spiking neurons. The categorization of baselines is confusing. Besides, the paper lacks details on the settings of these baselines.

**Questions:**

1. For those graphs following the power-law distribution, simply dividing nodes by their degrees may result in an uneven number of nodes between different groups. How does the grouping strategy proposed in the paper alleviate this issue?

2. In the experiment, the new and relevant spiking graph neural networks [2-3] and spiking neuron variants [4] can be introduced separately as baselines to demonstrate the model's effectiveness. Additionally, the authors should provide architectural details of baselines built from spiking neuron variants.

3. As mentioned in the paper, the real-world graphs tend to exhibit an extremely skewed distribution of degrees. The issue also remains in many large-scale graphs. Providing extra experiment results on node classification tasks would enhance the persuasiveness of the paper.

4. For spiking neural networks, there is a trade-off between the firing rate and the energy consumption. In this paper, the energy consumption of DAS-GNN is not only related to the firing rate but also to the grouping strategies. It would be clearer if the authors could provide the energy consumption (in formula) and further discuss the impact of the aforementioned settings on the energy consumption.

5. As depicted in Figure 5 and Appendix H, PGNN also shows highly diversified spike distributions on MUTAG and ENZYMES datasets. But MUTAG-PGNN-GCN and ENZYMES-PGNN-GAT remain obvious performance gaps compared to other baselines. It would be interesting if the authors could provide further discussions on these unexpected situations.

[1] Wei Fang et al. Incorporating learnable membrane time constant to enhance learning of spiking neural networks. In ICCV. 2021.

[2] Jintang Li et al. A Graph is Worth 1-bit Spikes: When Graph Contrastive Learning Meets Spiking Neural Networks. In ICLR. 2024.

[3] Mingkun Xu et al. Exploiting spiking dynamics with spatial-temporal feature normalization in graph learning. In IJCAI. 2021.

[4] Xingting Yao et al. GLIF: A Unified Gated Leaky Integrate-and-Fire Neuron for Spiking Neural Networks. In NIPS. 2022.

---

> ### Author Response · Authors · 2024-11-21
>
> We appreciate acknowledging the strength in our work and providing detailed feedback.
>
> ### **W1: Some descriptions of neurons and nodes are unclear. In Section 3, neurons are related to the feature dimension. But, in Section 4.2 the paper claims that neurons can be grouped by their degrees**
>
> $\to$ The neurons belonging to vertices of the same degree and the same feature position are grouped together. This yields groups shaped as vertically long bars as newly depicted with green boxes in Fig. 1(b) of the revised paper. We apologize for the confusion and moved the description regarding feature splitting to appear earlier in the revision.
>
> ### **W2: The definition of the “degree group” is vague and the paper lacks more explanations about the groups N_g or grouping strategies.**
>
> $\to$ A degree group comprises multiple degrees that can be found within an input graph. It is a concept we introduced in Appendix E to represent a case where we would want less number of groups than the number of unique degrees. We provided the definition in the revision.
> In addition, $N_g$ denotes the set of neurons within group $g$. We also clarified the definition in the revision.
>
>
>
>
>
> ### **W3: Confusion of categorizing baselines and lack of details on the settings of these baselines.**
>
> $\to$ All baselines are spiking neuron variants that are compared against our method at the neuron-level. To avoid confusion, we replaced their names in Table 1 with the type of neurons (e.g., Vanilla LIF neuron for SpikingGNN and adaptive techniques added neuron for SpikeNet). We apologize for the concern caused by the ambiguous terminology.
>
> We added experimental architecture details in the experiment settings on how we configured the baselines Appendix B and Section 5.1, such as the number of layers and hidden dimension size.
>
> ### **Q1: For those graphs following the power-law distribution, simply dividing nodes by their degrees may result in an uneven number of nodes between different groups. How does the grouping strategy proposed in the paper alleviate this issue?**
>
> $\to$ Yes, DAS-GNN could result in an uneven number of nodes between groups. However, we believe this would not be very problematic. Although the size of each group could be large, the neurons within the group will exhibit similar firing rates. Considering that the purpose of grouping is to share a threshold together among neurons with similar firing rates, we found that this empirically does not cause a problem. Please also see Table 7 for results with larger graphs.
>
> ### **Q2: In the experiment, the new and relevant spiking graph neural networks and spiking neuron variants can be introduced separately as baselines to demonstrate the model’s effectiveness. Additionally, the authors should provide architectural details of baselines built from spiking neuron variants.**
> |Method|MUTAG Orig|MUTAG DASGNN|PROTEINS Orig|PROTEINS DASGNN|ENZYMES Orig|ENZYMES DASGNN|NCI1 Orig|NCI1 DASGNN|IMDB-BINARY Orig|IMDB-BINARY DASGNN|
> |-|-:|:-|-:|:-|-:|:-|-:|:-|-:|:-|
> |SpikeGCL|91.49|97.34|77.87|79.15|28.17|32.17|65.69|67.76|71.70|73.30|
> |GC-SNN|88.33|93.13|72.33|73.32|43.50|53.00|63.36|65.38|69.90|77.20|
> |GA-SNN|66.49|92.05|59.57|65.85|33.17|51.83|52.21|66.13|50.00|78.60|
>
> $\to$ We added [4] to our main table (Table 1) as ‘GLIF’. The tendency of the result did not change after we added a new baseline (GLIF).
>
> Additionally, we conducted experiments with the three proposed techniques from [2-3], with and without techniques from DAS-GNN as shown in the table below. It shows that DAS-GNN is orthogonal to the techniques, and can be used to further improve the performance in all the tested cases. The result can be also found in Appendix K.
>
> In addition, we added architectural details of baselines on how we configured the baselines in Appendix B.

---

> ### Author Response · Authors · 2024-11-21
>
> ### **Q3: As mentioned in the paper, the real-world graphs tend to exhibit an extremely skewed distribution of degrees. The issue also remains in many large-scale graphs. Providing extra experiment results on node classification tasks would enhance the persuasiveness of the paper.**
>
> $\to$ We provide additional experiment results on the large-scale graphs for both graph classification and node classification tasks. We used REDDIT-BINARY and COLLAB datasets, which are large enough to exhibit a skewed distribution of degrees in the graph classification tasks. We show that DAS-GNN still outperforms when applied to large-scale graph classification tasks.
>
> In addition, we experimented with two node classification datasets: ogbn-arxiv, and ogbn-mag. DAS-GNN still performs the best, but the gap is smaller compared to the graph classification tasks. We believe this is because the node classification depends more on per-node properties rather than the graph structures, benefitting less from the proposed techniques.
>
> |Method|REDDIT-BINARY|COLLAB|OGBN-ARXIV|OGBN-MAG|
> |-|-|-|-|-|
> |**GCN**|||||
> |ANN|90.45|80.60|71.55|35.10|
> |SpikingGNN|81.10|67.24|53.52|27.78|
> |SpikeNet|82.60|68.78|60.18|28.82|
> |P-GNN|82.35|68.46|58.45|27.52|
> |GGNN|83.10|67.92|58.23|28.75|
> |ours|83.50|83.52|60.84|29.73|
> ||**0.40**|**14.74**|**0.66**|**0.91**|
> |**GAT**|||||
> |ANN|76.30|55.60|71.06|29.60|
> |SpikingGNN|50.00|52.00|52.21|14.05|
> |SpikeNet|50.05|52.00|53.67|17.30|
> |P-GNN|52.05|52.04|53.77|16.97|
> |GGNN|53.05|52.00|53.26|16.54|
> |ours|76.25|75.58|53.94|17.37|
> ||**23.20**|**23.54**|**0.17**|**0.07**|
> |**GIN**|||||
> |ANN|80.75|75.11|61.01|27.55|
> |SpikingGNN|79.75|52.04|56.72|23.31|
> |SpikeNet|79.70|53.66|55.79|23.37|
> |P-GNN|83.30|53.02|50.88|21.64|
> |GGNN|83.70|58.94|54.88|22.39|
> |ours|84.20|70.38|59.91|23.98|
> ||**0.50**|**11.44**|**3.19**|**0.61**|
>
> ### **Q4: DAS-GNN energy consumption is not only related to the firing but also to the grouping strategies. It would be clearer if the authors could provide the energy consumption (in the formula) and further discuss the impact of the aforementioned settings on the energy consumption.**
>
> $\to$ Yes, it is true that we need a small additional cost for updating the threshold per group.
> The overhead for computing the adaptive threshold operation would be $2 \cdot E_{AC} \times$ *num\_groups* $\times$ *hidden\_dimension* to account for the two additional multiplications related to $\gamma $per group in Eq. 10. We modified the energy consumption table to consider the operations needed for updating the thresholds, which demonstrates the small overhead. Please also find it in section 5.6 and Appendix J of the revision.
>
> ||MUTAG|PROTEINS|ENZYMES|NCI1|IMDB-BINARY|
> |-|-|-|-|-|-|
> |GCN|||||||
> |ANN|0.53mJ|6.92mJ|3.29mJ|21.41mJ|3.16mJ|
> |SpikingGNN|0.02mJ|1.28mJ|0.36mJ|3.92mJ|0.10mJ|
> |SpikeNet|0.06mJ|0.13mJ|0.25mJ|2.64mJ|0.35mJ|
> |PGNN|0.10mJ|0.79mJ|0.84mJ|6.54mJ|0.28mJ|
> |GGNN|0.08mJ|0.91mJ|0.48mJ|3.37mJ|0.21mJ|
> |+Additional cost|0.02μJ|0.08μJ|0.05μJ|0.02μJ|0.30μJ|
> |DAS-GNN|0.10mJ|0.94mJ|0.52mJ|5.28mJ|0.70mJ|
>
> ### **Q5: GNN also shows highly diversified spike distributions on MUTAG and ENZYMES datasets. But MUTAG-PGNN-GCN and ENZYMES-PGNN-GAT remain obvious performance gaps.**
>
> $\to$  We believe this is related to a trend we found where the spike rate diversity of the ultimate layer has an especially high correlation with the performance in general. In both the PGNN-MUTAG-GCN and PGNN-ENZYME-GAT, low diversity is observed ultimate layer (layer 3 for GCN, layer 2 for GAT). Although an exact reason is up for further investigation, we believe this is a reasonable behavior because it is directly used in the final output. We have discussed this in Section 5.5 and Figures 27-28 of the revision.

---

> > ### Comment · Reviewer_skyv · 2024-11-27
> > **Reply to the Authors**
> >
> > I would like to thank the authors for their response. The above response has resolved some of my concerns, and we are willing to raise my rating. However, we believe the current method still has several limitations:
> >
> > 1. As mentioned in the response, the current grouping strategy will result in uneven numbers of nodes in each group. Therefore,   allowing low-degree nodes to have higher firing rates seems to be a double-edged sword. For large-scale graphs, a large number of nodes with the low degree generate numerous spikes would undoubtedly increase the energy consumption of the model. A more in-depth exploration of the grouping strategy is necessary, and the paper fails to delve into this intriguing issue.
> >
> > 2. For empirical experiments, many advanced spiking neuron variants have been proposed including learnable threshold voltages and temporal-based encoding mechanisms. The baselines like LIF and Adaptive LIF are quite simple and outdated, making it difficult to determine whether the proposed methods have a significant improvement in enhancing spike diversity compared to other advanced neuron variants.
> >
> > 3. The typical use cases provided in the paper do not involve node features, which limits the scalability of the proposed neurons.

---

> > > ### Author Response · Authors · 2024-11-28
> > >
> > > Thank you for taking the time to share your thoughts. We genuinely appreciate your comments and feedback, and we're confident your insights will help us further enhance our work.
> > >
> > > If you have any other questions or comments, please don’t hesitate to reach out to us at any time.

---

### Official Review · Reviewer_ApnA · 2024-11-02

**Soundness:** 3
**Presentation:** 3
**Contribution:** 3
**Rating:** 6
**Confidence:** 4

**Summary:**

This paper proposes degree-aware spiking graph neural networks with adaptive thresholds based on a group of neurons for graph classification. The paper first diagnoses the poor performance as the existence of neurons under starvation caused by the graph structure. Then the paper proposes adaptive threshold among neurons partitioned by degrees, as well as learnable initial threshold and decay rate to reduce the sensitivity. Experiments on several datasets show superior performance of the proposed method and the potential low energy costs.

**Strengths:**

1. This paper identifies the starvation problem of spiking graph neural networks that causes performance drop when adapting them to graph classification.

2. This paper proposes a novel degree-aware group-adaptive technique to overcome the problem.

3. Experiments show superior performance on several datasets, some outperforming ANNs.

**Weaknesses:**

1. This paper does not discuss the influence of the proposed method for deployment on potential neuromorphic hardware, while SNNs mainly target those hardware to obtain energy efficiency. The proposed degree-aware group-adaptive neurons require the thresholds to depend on other neurons. Is it plausible for potential neuromorphic hardware? Or will it introduce much more computation for communications between neurons?

2. It is not clear if the energy consumption analysis takes the costs of this adaptive threshold operation (and potential communications between neurons) into account.

**Questions:**

Some recent works also study SNN for link prediction tasks in graphs besides node-level classification [1], which aims to better leverage the temporal spiking time property of SNNs beyond just the spiking nature. For the proposed method, is only spike rate considered? Is it possible to better leverage these properties of SNNs beyond rate, which are considered important features since the origin of SNNs [2]? These can be discussed.

[1] Temporal Spiking Neural Networks with Synaptic Delay for Graph Reasoning. ICML 2024.

[2] Networks of spiking neurons: the third generation of neural network models. Neural Networks, 1997.

---

> ### Author Response · Authors · 2024-11-21
>
> We thank the reviewer for acknowledging the novelty of the work and providing constructive feedback. We have addressed the comments below.
> ### **W1: The proposed degree-aware group-adaptive neurons require the thresholds to depend on other neurons. Is it plausible for potential neuromorphic hardware? Or will it introduce much more computation for communications between neurons?.**
>
> $\to$ Yes. Our method would indeed be plausible for potential neuromorphic hardware [1-3] without introducing significant computational overhead for communication between neurons.
> Neuromorphic processors are typically designed with neuron cores as fundamental units, where each core handles neuron states (e.g., membrane potential) stored in local memory (often an SRAM), updates them based on synaptic inputs, and generates spikes upon reaching a threshold.
> To implement DAS-GNN, each neuron group can be placed together in a core as a whole. Then, the sum of firing rates (Eq. 9) can reside in the local memory and be updated by counting spikes from each neuron of the same group.  Since the threshold adjustment happens locally at the core level, it doesn’t need extensive communication overhead between neurons. Please find this discussion in section 5.6 of the revision.
>
> [1]  Merolla, Paul A., et al. "A million spiking-neuron integrated circuit with a scalable communication network and interface." Science 345.6197 (2014): 668-673.
>
> [2] Akopyan, Filipp, et al. "Truenorth: Design and tool flow of a 65 mw 1 million neuron programmable neurosynaptic chip." IEEE transactions on computer-aided design of integrated circuits and systems 34.10 (2015): 1537-1557.
>
> [3] Davies, Mike, et al. "Loihi: A neuromorphic manycore processor with on-chip learning." Ieee Micro 38.1 (2018): 82-99.
>
> ### **W2: It is not clear if the energy consumption analysis takes the costs of this adaptive threshold operation (and potential communications between neurons) into account.**
>
> $\to$ We updated the energy consumption analysis to include the adaptive threshold operation. As mentioned in W1, DAS-GNN does not incur any additional communication.
> The overhead for computing the adaptive threshold operation would be $2 \cdot E_{AC} \times$ *num\_groups* $\times$ *hidden\_dimension* to account for the two additional multiplications related to $\gamma $per group in Eq.10. However, as shown in the table, this only adds a minimal amount of overhead to the energy consumption. This was also added to section 5.6 of the revised version.
>
> ||MUTAG|PROTEINS|ENZYMES|NCI1|IMDB-BINARY|
> |-|-|-|-|-|-|
> |GCN|||||||
> |ANN|0.53mJ|6.92mJ|3.29mJ|21.41mJ|3.16mJ|
> |SpikingGNN|0.02mJ|1.28mJ|0.36mJ|3.92mJ|0.10mJ|
> |SpikeNet|0.06mJ|0.13mJ|0.25mJ|2.64mJ|0.35mJ|
> |PGNN|0.10mJ|0.79mJ|0.84mJ|6.54mJ|0.28mJ|
> |GGNN|0.08mJ|0.91mJ|0.48mJ|3.37mJ|0.21mJ|
> |+Additional cost|0.02μJ|0.08μJ|0.05μJ|0.02μJ|0.30μJ|
> |DAS-GNN|0.10mJ|0.94mJ|0.52mJ|5.28mJ|0.70mJ|
>
>
>
>
> ### **Q1. For the proposed method, is only the spike rate considered?**
> $\to$ We agree that considering temporal information could further enhance the performance of DAS-GNN. For example, we could apply a similar idea by assigning synaptic delays based on the degree of each node, or encode community information as a temporal property. We added the discussion in Appendix K.

---

> > ### Comment · Reviewer_ApnA · 2024-11-26
> >
> > I thank the authors for the responses. I have an additional question regarding the GGNN results supplemented in the energy comparison Table in the response. Why does it show significantly smaller costs than DAS-GNN? The percentage of its energy saving compared with DAS-GNN is larger than that of DAS-GNN compared with ANN. From Table 1 in the revised version, its performance is also acceptable. This may influence the claim "DAS-GNN maintains energy consumption comparable to other baseline SNN models". Can the authors provide more discussions?

---

> > > ### Author Response · Authors · 2024-11-26
> > >
> > > We sincerely apologize for our mistake in the rebuttal table regarding the GGNN values. We found there was a corruption during adding a new baseline to our automated formula. We replaced the table with the corrected values. Looking at the correct values, GGNN’s energy consumption shows similar trends to those of the other SNN baselines’ energy consumption values.

---

> > > > ### Comment · Reviewer_ApnA · 2024-11-27
> > > >
> > > > Thank you for the clarification and I'm glad to keep the positive rating.

---

> > > > > ### Author Response · Authors · 2024-11-28
> > > > >
> > > > > Thank you! We greatly appreciate the reviewer’s thorough consideration of our response and the valuable insights shared with us.
> > > > > If you have any additional questions or thoughts, please feel free to contact us.

---

### Official Review · Reviewer_bkyi · 2024-11-04

**Soundness:** 3
**Presentation:** 3
**Contribution:** 3
**Rating:** 8
**Confidence:** 4

**Summary:**

The article introduces DAS-GNN, a novel approach for graph classification task using spiking neural networks. The paper discusses the challenges of applying SNNs for graph classification i.e. varying spike frequency and to overcome these challenge authors proposed a Degree aware group adaptive neurons (DAG) that group neurons based on node degrees, and a Learnable inference base thresholds (LIBT) to reduce sensitivity of DAG to inference thresholds, The proposed method shows significant improvements over baseline approaches and even outperforms traditional ANNs in several cases while maintaining the energy efficiency.

**Strengths:**

1.In-depth Analysis of Spike Frequency Variation: Significant strength of the work is that authors have thoroughly studied spike frequency variation problem in graph networks along with providing clear visualizations which makes its easier to follow.
2- Ablation studies : The authors conducted component wise ablation studies which helps to understand the method contributing most to performance improvements. The authors have also conducted additional experiments to understand sensitivity to different hyperparameters such as threshold and learning rates.

**Weaknesses:**

1 - The paper lacks the theoretical justification of why the proposed DAG method provides better performance than other methods.
2-  Simplifying mathematical notations and providing detailed explanations for degree-aware neuron adaptation might make it more accessible to wider audience.Also the model architecture and spiking mechanism needs to be more detailed.

**Questions:**

1- How would the proposed method scale to very large graph with very skewed degree distributions?
2- Did authors perform any sensitivity for any other hyperparameters?

---

> ### Author Response · Authors · 2024-11-21
>
> We thank the reviewer for acknowledging our contributions and giving positive feedback. We try our best to faithfully address the comments below.
>
> ### **W1: Lack of the theoretical justification of the DAG method.**
> $\to$ Although a complete analysis would be our future work, we provide a rough theoretical justification for grouping neurons based on their degree.
> By reformulating the message passing part of Eq.7 in the perspective of a single neuron, the membrane potential of a neuron $i$ is computed as $U_i=\sum_{j\in active(i)}W_{i,j}$, where $active(i)$ denotes firing neurons among neighbors of $i$. When we follow a common assumption where $W_{i,j} \sim N(0,\sigma)$, it leads to $U_i \sim N(0, \sigma \cdot |active(i)|)$. Since $U_i$ determines the firing rate and $|active(i)|$ is proportional to the degree of $i$, grouping neurons by its vertex degree has an effect of grouping neurons with similar membrane potential variance. This aligns with our findings from Section 3, and we added this analysis in Section 4.2 of the revised paper.
> ### **W2: Simplifying mathematical notations and providing detailed explanations for spike mechanism and model architecture.**
> $\to$ We simplified some mathematical notations in Eqs.8-10 and clarified explanations. In addition, we added model architecture details for experimental settings.
> A major change is that we changed the symbol for membrane potential from $V(t)$ to $U(t)$, which could have caused some confusion with the threshold voltage $V_{th}$.
> In addition, we added our proposed grouping method in Figure 1(b) with green boxes for more intuitive understanding.
> ### **Q1. Scalability to a very large graph with skewed degree distribution**
> $\to$ We show additional experiment results on two large-scale graph datasets used for graph classification: REDDIT-BINARY and COLLAB. We could find that our DAS-GNN still outperforms other baselines in large graphs with skewed degree distribution. Please also find these results in Appendix D of the revision.
> For another demonstration on performance on large graphs, we extend our method to node classification datasets (ogbn-arxiv, ogbn-mag), where the results further show that DAS-GNN maintains its competitive performance.
>
> |Method|REDDIT-BINARY|COLLAB|OGBN-ARXIV|OGBN-MAG|
> |-|-|-|-|-|
> |**GCN**|||||
> |ANN|90.45|80.60|71.55|35.10|
> |SpikingGNN|81.10|67.24|53.52|27.78|
> |SpikeNet|82.60|68.78|60.18|28.82|
> |PGNN|82.35|68.46|58.45|27.52|
> |GGNN|83.10|67.92|58.23|28.75|
> |Ours|83.50|83.52|60.84|29.73|
> ||**0.40**|**14.74**|**0.66**|**0.91**|
> |**GAT**|||||
> |ANN|76.30|55.60|71.06|29.60|
> |SpikingGNN|50.00|52.00|52.21|14.05|
> |SpikeNet|50.05|52.00|53.67|17.30|
> |PGNN|52.05|52.04|53.77|16.97|
> |GGNN|53.05|52.00|53.26|16.54|
> |Ours|76.25|75.58|53.94|17.37|
> ||**23.20**|**23.54**|**0.17**|**0.07**|
> |**GIN**|||||
> |ANN|80.75|75.11|61.01|27.55|
> |SpikingGNN|79.75|52.04|56.72|23.31|
> |SpikeNet|79.70|53.66|55.79|23.37|
> |PGNN|83.30|53.02|50.88|21.64|
> |GGNN|83.70|58.94|54.88|22.39|
> |Ours|84.20|70.38|59.91|23.98|
> ||**0.50**|**11.44**|**3.19**|**0.61**|

---

> ### Author Response · Authors · 2024-11-21
>
> ### **Q2. Sensitivity for other parameters**
>
> $\to$ We provide additional sensitivity studies regarding $\gamma$ from Eq. 10, and the size of GNN hidden dimensions.
> We find that DAG is generally insensitive to $\gamma$, with minimal degradation within the given range. We use 0.2 as the default value, which is generally the best setting across different model architectures.
> For the study on GNN hidden dimension size, we find that our method is insensitive to varying sizes of GNN, consistently outperforming PLIF neurons across different hidden dimension sizes.
> Please also find the results in Table 3. and Appendix H, Appendix I of the revision.
>
> **Table: Sensitivity study on $\gamma$**
> |$\gamma$||0.05|0.10|0.15|0.20|0.25|0.30|0.35|0.40|
> |-|-|-:|-:|-:|-:|-:|-:|-:|-:|
> |MUTAG|GCN|95.79|96.84|96.32|97.37|93.13|96.32|96.29|95.76|
> ||GAT|95.23|93.66|92.60|94.21|94.74|92.60|92.60|92.08|
> ||GIN|96.84|95.26|95.79|96.32|95.26|95.26|95.26|95.79|
> |PROTEINS|GCN|76.64|77.09|77.36|77.72|76.91|77.18|77.36|77.09|
> ||GAT|70.62|70.53|71.70|72.14|71.70|74.40|72.60|71.34|
> ||GIN|79.51|79.78|79.52|80.02|79.78|79.15|79.42|78.98|
> |ENZYMES|GCN|63.17|61.50|61.33|60.17|61.67|60.83|61.17|62.33|
> ||GAT|60.67|61.67|61.33|61.00|62.83|60.83|62.83|63.33|
> ||GIN|55.50|54.50|54.00|57.83|56.50|57.00|51.83|50.83|
> |NCI1|GCN|76.91|76.62|76.59|77.25|76.52|76.50|75.01|74.60|
> ||GAT|74.70|74.96|74.14|73.82|74.31|73.82|73.09|73.33|
> ||GIN|78.61|77.47|77.25|77.45|75.64|75.13|75.04|73.41|
> |IMDB-B|GCN|80.70|80.10|80.40|80.60|80.40|80.80|81.00|80.50|
> ||GAT|75.40|75.80|76.60|77.80|78.20|77.10|77.10|78.10|
> ||GIN|76.70|78.00|77.70|79.40|78.70|79.00|78.40|78.10|
>
>
>
> **Table: Sensitivity on layer hidden dimension DAS-GNN**
> |Hidden Dim.||32|64|128|256|
> |-|-|-:|-:|-:|-:|
> |MUTAG|GCN|94.74|95.79|97.37|96.84|
> ||GAT|94.74|94.21|93.68|93.68|
> ||GIN|95.79|95.76|96.32|93.65|
> |PROTEINS|GCN|76.64|77.99|77.72|77.99|
> ||GAT|73.58|72.14|69.91|70.23|
> ||GIN|80.14|79.33|80.02|79.70|
> |ENZYMES|GCN|53.33|56.50|60.17|64.00|
> ||GAT|58.33|61.00|54.83|55.83|
> ||GIN|53.67|57.33|57.83|57.33|
> |NCI1|GCN|72.19|74.09|77.25|78.13|
> ||GAT|75.88|73.82|75.16|75.72|
> ||GIN|76.81|76.11|77.45|75.67|
> |IMDB-B|GCN|80.30|80.30|80.60|80.90|
> ||GAT|77.00|77.80|76.70|77.30|
> ||GIN|80.00|78.90|79.40|76.20|
>
>
>
> **Table: Sensitivity on layer hidden dimension PLIF**
> |Hidden Dim.||32|64|128|256|
> |-|-|-:|-:|-:|-:|
> |MUTAG|GCN|87.81|86.75|87.28|88.33|
> ||GAT|83.01|82.49|83.01|83.01|
> ||GIN|93.13|93.13|94.18|94.18|
> |PROTEINS|GCN|75.11|76.82|77.72|76.46|
> ||GAT|68.73|64.06|67.29|66.93|
> ||GIN|79.07|78.44|79.16|77.81|
> |ENZYMES|GCN|53.00|57.17|60.50|59.83|
> ||GAT|37.17|38.50|37.83|36.67|
> ||GIN|51.83|51.50|50.17|48.67|
> |NCI1|GCN|70.39|72.60|76.52|77.69|
> ||GAT|60.40|68.32|60.42|58.52|
> ||GIN|73.70|73.33|75.38|72.02|
> |IMDB-B|GCN|66.80|68.80|71.60|71.30|
> ||GAT|50.00|50.00|50.00|50.00|
> ||GIN|74.90|76.30|72.80|69.80|

---

### Author Response · Authors · 2024-11-21
**General Response**

We sincerely thank all the reviewers for giving us insightful, valuable comments which have helped us improve our work.

We have revised our manuscript and the changes can be summarized as follows:

* Additional baselines for the main table in the experiment section (Table 1, GLIF [1]).
* Modification of energy consumption analysis to consider adaptive operations.
* Additional experiments on large-scale graph classification datasets (REDDIT-BINARY, COLLAB) in Appendix D.
* Additional sensitivity study regarding adaptive step size and hidden dimension.
* Additional description of the spiking mechanism.
* Additional architectural details for our baselines.


[1]  Xingting Yao et al. GLIF: A Unified Gated Leaky Integrate-and-Fire Neuron for Spiking Neural Networks. In NIPS. 2022.

---

### Meta-Review · Area_Chair_FTzY · 2024-12-22

**Metareview:**

Summary: DAS-GNN is a novel approach for graph classification using spiking neural networks (SNNs) that addresses the challenge of varying spike frequency by introducing Degree-aware Group Adaptive Neurons (DAG) and Learnable Inference Base Thresholds (LIBT). The method demonstrates significant improvements over baseline approaches and even outperforms traditional ANNs in some cases, while maintaining energy efficiency.

Strengths:

DAS-GNN provides an in-depth analysis of spike frequency variation in graph networks and offers a thorough understanding of the problem through clear visualizations and ablation studies.

The method shows superior performance on several datasets, outperforming ANNs and highlighting its potential for energy-saving and accurate graph-based systems.

Weaknesses:

The paper lacks a theoretical justification for the proposed DAG method and could benefit from simplifying mathematical notations and providing more detailed explanations of the degree-aware neuron adaptation.

There is a lack of discussion on the influence of the proposed method for deployment on neuromorphic hardware, and the energy consumption analysis may not fully account for the costs of the adaptive threshold operation and potential communications between neurons.

Given the mixed results and the concerns raised about the paper's clarity, theoretical justification, and practical implications for neuromorphic hardware, I must reject this work as it does not fully meet the acceptance criteria.

**Additional Comments On Reviewer Discussion:**

Concerns are not well-addressed.

---

### Decision · Program_Chairs · 2025-01-22

Reject